# RISE: Robust Individualized Decision Learning with Sensitive Variables

**Xiaoqing Tan**
University of Pittsburgh
xit31@pitt.edu

**Zhengling Qi**
George Washington University
qizhengling@email.gwu.edu

**Christopher W. Seymour**
University of Pittsburgh
seymourc@pitt.edu

**Lu Tang**[*]
University of Pittsburgh
lutang@pitt.edu

## Abstract

This paper introduces RISE, a robust individualized decision learning framework with sensitive variables, where sensitive variables are collectible data and important to the intervention decision, but their inclusion in decision making is prohibited due to reasons such as delayed availability or fairness concerns. A naive baseline is to ignore these sensitive variables in learning decision rules, leading to significant uncertainty and bias. To address this, we propose a decision learning framework to incorporate sensitive variables during *offline* training but not include them in the input of the learned decision rule during model deployment. Specifically, from a causal perspective, the proposed framework intends to improve the worst-case outcomes of individuals caused by sensitive variables that are unavailable at the time of decision. Unlike most existing literature that uses mean-optimal objectives, we propose a robust learning framework by finding a newly defined quantile- or infimum-optimal decision rule. The reliable performance of the proposed method is demonstrated through synthetic experiments and three real-world applications.

## 1 Introduction

Recently, there has been a widespread interest in developing methodology for individualized decision rules (IDRs) based on observational data. When deriving IDRs, some collectible data are important to the intervention decision, while their inclusion in decision making is prohibited due to reasons such as delayed availability or fairness concerns. For example, sensitive characteristics of subjects regarding their income, sex, race and ethnicity may not be appropriate to be used directly for decision making due to fairness concerns. In the medical field especially for patients in severe life-threatening conditions such as sepsis, timely bedside intervention decisions have to be made before lab measurements are ordered, assayed and returned to the attending physicians. However, due to the delayed availability of lab results, most of the decisions are made with great uncertainty and bias due to partial information at hand. We define *sensitive variables* as variables whose inclusion into decision rules is prohibited. The formal definition of sensitive variables will be given in Section 3.

In this work, we propose RISE (**R**obust **I**ndividualized decision learning with **SE**nsitive variables)[1], a robust IDR framework to improve the outcome of individuals when there are informative yet sensitive variables that are either not available or prohibited from using during IDR deployment. To achieve this, we propose to estimate the optimal IDR by optimizing a quantile- or infimum-based objective,

---

[*]Corresponding author.
[1]Python code is available at https://github.com/ellenxtan/rise.

36th Conference on Neural Information Processing Systems (NeurIPS 2022).

respectively, for continuous or discrete sensitive variables. This optimization problem is then shown to be equivalent to a weighted classification problem where *most existing machine learning classifiers can be readily applied*. Our idea falls along the lines of work that considers algorithmic fairness [15] while extending it to the setting of causal inference [56] in the sense that decisions are driven by causality rather than a general utility function. We show in our empirical analyses that this leads to fairer and safer real-life decisions with little sacrifice of the overall performance.

Assuming that a larger outcome value is preferable, optimal IDRs are traditionally derived through maximizing the mean outcome of the sample population. In this paper, we are interested in a specific yet broadly applicable setting of learning that involves sensitive variables. We consider offline learning where sensitive variables are collected and *can be used in training the IDRs*, but they *cannot be used as input in the resulting IDRs*. This is a setting commonly considered in the fairness and privacy-related literature for classification (e.g., [23]), but not from a causal standpoint. When there exist important variables that are simply left out from training, the estimated IDR will be biased. This bias can be removed if all important variables are used during training, which we will show in Section 3.1 a mean-optimal approach. The optimal action maximizes the mean outcome where the mean is taken over the sensitive variables, conditioning on other variables. This method, however, has no control of the disparity in sensitive variables. Subjects with different sensitive values may report large outcome differences, hence unfairly or unsafely treated. Therefore, objective functions with robustness guarantees for sensitive variables are preferred, since they offer protection to subjects in the lower tail of the outcome distribution with regards to the sensitive values.

For illustration, we consider a toy example with binary actions, $A \in \{-1, 1\}$. We remark that the decision can only be made based on the variable $X$ whereas $S$ is a sensitive variable. The setup is shown in Table 1 and the oracle values under the mean-optimal rule and RISE[2] are given in Table 2. The detailed setup can be found in Section 4.1 under Example 1. We consider *vulnerable subjects* as those with low outcome values, as highlighted in red in Table 1 (A full definition is given in Section 3.2). For $X \leq 0.5$, the mean-optimal rule would assign action $A = 1$ as it tries to achieve the largest average reward across $S = 0$ and $S = 1$. Recall that $S$ is not available at the time of decision. However, this action results in great harm for subjects with $S = 1$ as they could get the worst expected outcome of 0. On the contrary, RISE improves the worst-case outcome by assigning $A = -1$, protecting the vulnerable subjects. Likewise, for $X > 0.5$, the mean-optimal rule assigns $A = -1$ while RISE assigns $A = 1$ protecting those with $S = 0$ that could have experienced an outcome of 5. Compared to the mean-optimal rule, the proposed rule achieves a larger reward among vulnerable subjects while maintaining a comparable overall reward.

Table 1: Toy example setup of $E(Y|X, S, A)$.

|  | $X \leq 0.5$ | | $X > 0.5$ | |
| --- | --- | --- | --- | --- |
|  | $S = 0$ | $S = 1$ | $S = 0$ | $S = 1$ |
| $A = -1$ | 11 | 13 | 5 | 27 |
| $A = 1$ | 30 | 0 | 15 | 13 |

Table 2: Toy example results.

|  | Average reward | |
| --- | --- | --- |
|  | Overall | Vulnerable |
| Mean-optimal rule | **15.5** | 2.5 |
| RISE | 13 | **14** |

**Main contributions.** *Methodology-wise, 1)* we propose a novel framework, RISE, to handle sensitive variables in causality-driven decision making. Robustness is introduced to improve the worst-case outcome caused by sensitive variables, and as a result, it reduces the outcome variation across subjects. The latter is directly associated with fairness and safety in decision making. To the best of our knowledge, we are among the first to propose a robust-type fairness criterion under causal inference. *2)* We introduce a classification-based optimization framework that can easily leverage most existing classification tools catered to different functional classes, including state-of-the-art random forest, boosting, or neural network models. *Application-wise, 3)* the consideration of sensitive variables in decision learning is important to applications in policy, education, healthcare, etc. Specifically, we illustrate the application of RISE using three real-world examples from fairness and safety perspectives where robust decision rules are needed, across which we have observed robust performance of the proposed approach. From a fairness perspective, we consider a job training program where age is considered as a sensitive variable. From a safety perspective, we consider two applications to healthcare where lab measurements are considered as sensitive variables.

---

[2]For the mean-optimal rule, overall average reward is calculated by $(30 + 0 + 5 + 27)/4 = 15.5$, reward among vulnerable subjects is calculated by $(0 + 5)/2 = 2.5$; for RISE, overall average reward is calculated by $(11 + 13 + 15 + 13)/4 = 13$, reward among vulnerable subjects is calculated by $(13 + 15)/2 = 14$.

## 2 Related Work

Our work focuses on individualized decision rules, which aim at assigning treatment decision based on subject characteristics. Existing methods for deriving IDRs include model-based methods such as Q-learning [75, 39, 38] and A-learning [53, 61], model-free policy search methods [77, 81, 80], and contextual bandit methods [6, 31]. In Appendix A, we provide additional literature review on general IDRs under causal settings. Fairness, safety and robustness are topics of interest that extend well beyond causal inference. In the following, we provide a review of these areas, with focus given to work related to causal inference and IDRs.

**Fairness and safety in IDRs.** The consideration of fairness and safety in machine learning has seen an explosion of interest in the past few years [15, 69, 4, 43, 20, 12, 37, 47, 65, 66], especially for solving classification and regression problems to help derive decisions that are not only accurate but also fair. In these work, sensitive variables are also referred to as sensitive, protected, or auxiliary attributes. We extend the definition of sensitive variables to include delayed information that is not available at deployment as it is also suitable for this framework.

Among earlier work, preprocessing [23, 17, 13, 58] and inprocess training approaches [5, 20, 29] consider disentangling the input $X$ from a known or unknown sensitive variable $S$ so that the transformed $X$ does not contain any information related to $S$. Due to the causal nature of IDRs, effect of IDRs cannot be estimated consistently when an informative $S$ is left out and the resulting rule is suboptimal. This follows from the classic argument that any unmeasured confounding (i.e., $S$), if not accounted for, would lead to bias. Similar issues persist in contextual bandits [22, 46]. [35] considers reducing the impact of auxiliary variables on prediction under distributional shift. Although it is motivated from a causal idea, its main focus is still on prediction. Inside the causal framework, [79, 42] extend fairness from prediction to policy learning using causal graphical models by incorporating fairness constraints. [10] considers counterfactual fairness that seeks to achieve conditional independence of the decisions via data preprocessing. Despite earlier efforts in bringing fairness into the causal framework, most of these approaches only ensure mean zero disparity in $S$ but do not have robustness guarantees in the sense that the variance of the disparity in $S$ is not controlled. Besides, most examples consider a single categorical sensitive variable, but not multiple or continuous ones.

**Robustness in IDRs.** Recently the statistical literature has witnessed a growing interest in developing robust methods for estimating IDRs. They introduce robustness into the objective function by using quantile-optimal treatment regimes or mean-optimal treatment regimes under certain constraints to improve the gain of individuals at the lower tail of the reward spectrum [72, 74, 51, 49, 16]. In particular, [72, 51] propose to estimate quantile- or tail-optimal treatment regimes. [74] studies the mean-optimal treatment regime under a constraint to control for the average potential risk. [49] proposes a decision-rule-based optimized covariates dependent equivalent for tasks of individualized decision making. [16] considers mean and quantile objectives simultaneously by maximizing the average value with the guarantee that its tail performance exceeds a prespecified threshold. Robustness, in their sense, pertains to the outcome distribution subject to the sampling error. When sensitive variables are present, we consider instead the robustness of the outcome distribution subject to the uncertainty due to sensitive variables, providing a more targeted way of ensuring robustness, which is directly related to fairness and safety. Compared to algorithms based on explicit fairness constraints (for example [76, 78] in classification and [79, 10] in causal inference) that seek to remove the disparity across different values of $S$, our method reduces the variance of disparity across $S$. In addition, constraint-based approaches typically require specialized optimization procedures whereas our approach presents an elegant and systematic way for optimization. To our knowledge, we are the first few to consider decision fairness via a robust objective under the causal framework.

## 3 Robust Decision Learning Framework with Sensitive Variables

### 3.1 Preliminaries

**Notation.** We let random variables be represented by upper-case letters, and their realizations be represented by lower-case letters. Suppose there are $n$ independent subjects sampled from a given population. For subject $i$, let $A_i \in \{-1, 1\}$ denote a binary treatment assignment and $Y_i$ denote the corresponding outcome. Without loss of generality, we assume a larger value of outcome is desirable.

Under the potential outcomes framework [55, 63], let $Y_i(a)$ be the potential outcome had the subject been assigned to $A = a$ for $a = 1$ or $-1$. Let $X_i \in \mathbb{X}$ be the feature vector and, for now, $S_i$ be a single sensitive variable. We consider $S \in \mathbb{S}$ where $\mathbb{S} = \{1, \ldots, K\}$ if $S$ is discrete and $\mathbb{S} = \mathbb{R}$ if $S$ is continuous. The extension to multiple sensitive variables is presented in Section 3.4.

**Definition of sensitive variables.** We define sensitive variables $S$ as variables that are important to the intervention decision, but their inclusion in decision making is prohibited. Formally, consider variables $X$ and $S$ that are both available during model training and are both determinants of conditional average treatment effect [54]. While $S$ may be involved in training, the derived decision rule $d(\cdot)$ precludes the input of $S$ due to sensitive concerns. Hence, the derived IDR is a function of the form $d(X) : \mathbb{X} \mapsto \mathbb{A}$. Following the above definition, we consider an offline learning framework where sensitive variables are collected and can be used in obtaining the IDRs, but they cannot be used in the resulting IDRs. A causal diagram and a decision diagram are provided in Figure 1. As it shows in Figure 1a, both $X$ and $S$ confound the effect of treatment $A$ on outcome $Y$. The arrows represent the causal relationship between variables. Note that $X$ and $S$ can be correlated. This causal diagram is formalized in Assumption 1 below. On the other hand in Figure 1b, in the decision diagram under our setting, $S$ is shown in a dotted circle as $S$ may not be readily available at the time of decision making. We connect $S$ and $A$ with a dotted arrow to indicate that $S$ is incorporated in the training of the decision rule, but it is not required at deployment.

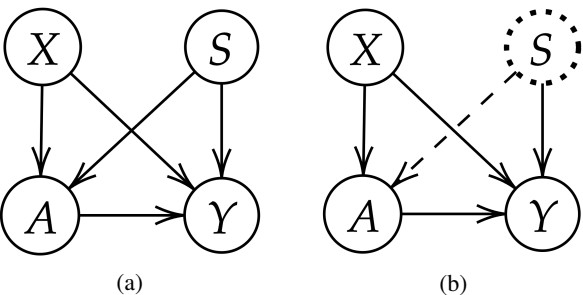

Figure 1: (a) A causal diagram. (b) A decision diagram.

**Assumption 1.** *Assume the following conditions hold:*

*1a Consistency:* $Y = Y(-1)\mathbb{1}(A = -1) + Y(1)\mathbb{1}(A = 1)$.

*1b Positivity:* $0 < Pr(A = 1|X, S) < 1$.

*1c Unconfoundedness:* $\{Y(-1), Y(1)\} \perp A|\{X, S\}$ *and* $\{Y(-1), Y(1)\} \not\perp A|X$.

Assumption 1a is the standard consistency assumption in causal inference and Assumption 1b states that every subject has a nonzero probability of getting the treatment. Assumption 1c states that given $X$ and $S$, the potential outcomes are independent of the treatment assignments. Besides, unconfoundedness does not hold when only $X$ is given, signifying the important role of $S$. Under causal settings, Assumption 1c implies that treatment effects cannot be non-parametrically identifiable without $S$ [44, 54]. Approaches such as ones that disentangle $X$ from $S$ under supervised learning settings mentioned in Section 2 will introduce bias towards estimating IDRs.

**Naive approaches that omit sensitive variables.** When $S$ is not available for future deployment, a naive approach is to maximize $E_X\{E(Y|X, A = d(X))\}$ over $d$ using $(X, A, Y)$ during the training procedure. This approach introduces bias in the estimation of conditional treatment effect and leads to a suboptimal IDR due to the unmeasured confounder $S$.

**Mean-optimal approaches that use the sensitive variables.** It is thus important that one includes $S$ into the training procedure. For example, if we consider the value function framework (i.e., expected outcome) used by most existing works such as [36, 52], we can show that

$$E\{Y(d)\} = E_{X,S}\big[E(Y(d)|X, S)\big] = E_X\big[E_{S|X}\{E(Y(d)|X, S)\}\big] \tag{1}$$
$$= E_X\big[E_{S|X}\{E(Y|X, S, A = d(X))\}\big] \neq E_X\big[E(Y|X, A = d(X))\big],$$

where the third equality in (1) holds by Assumption 1 and the last inequality also indicates the naive approaches without using $S$ will in general fail. Then one valid approach is to maximize

$E_X\big[E_{S|X}\{E(Y|X,S,A=d(X))\}\big]$ over $d$ using $(X,S,A,Y)$. The optimal IDR under this criterion is, for every $X \in \mathbb{X}$,

$$\tilde{d}(X) \in \text{sgn}(E_{S|X}\{E(Y|X,S,A=1)\} - E_{S|X}\{E(Y|X,S,A=-1)\}),$$

which finds the treatment that maximizes the conditional expected outcome given $X$ by averaging out the effect of the sensitive variable $S$. Mean-optimal approaches, however, fail to control the disparities across realizations of the sensitive variables due to the integration over $S$, which may lead to unsatisfactory decisions to certain subgroups, as illustrated in the toy example in Section 1.

## 3.2 Robust Optimality with Sensitive Variables

Driven by the limitation of existing approaches, our goal is to derive a robust decision rule that maximizes the worst-case scenarios of subjects when some sensitive information is not available at the time of deploying the decision rule. Specifically, our robust decision learning framework draws decisions based on individuals' available characteristics summarized in the vector $X$ without the sensitive variable $S$, while improving the worst-case outcome of subjects in terms of the sensitive variable in the population. Formally, given a collection $\mathbb{D}$ of all treatment decision rules depending only on $X$, the proposed RISE approach estimates the following IDR, which is defined as

$$d^* \in \arg\max_{d \in \mathbb{D}} E_X\big[G_{S|X}\{E(Y|X,S,A=d(X))\}\big], \qquad (2)$$

where $G_{S|X}(\cdot)$ could be chosen as some risk measure for evaluating $E(Y|X,S,A=d(X))$ for each $S \in \mathbb{S}$. Examples include variance, conditional value at risk, quantiles, etc. In this paper, we consider $G_{S|X}$ as the conditional quantiles (for a continuous $S$) or the infimum (for a discrete $S$) over $\mathbb{S}$.

Specifically, for a discrete $S$, $G_{S|X}$ is considered as an infimum operator of $E(Y|X,S,A=d(X))$ over $S$. We thus aim to find

$$d^* \in \arg\max_{d \in \mathbb{D}} E_X\big[\inf_{s \in \mathbb{S}}\{E(Y|X,S=s,A=d(X))\}\big],$$

where $\inf$ is the infimum taken with respect to $E(Y|X,s,A=d(X))$ over $s \in \mathbb{S}$. This implies that for a given $X$, $d^*(X)$ assigns the treatment that yields the best worst-case scenario among all possible values of $S$ for every $X \in \mathbb{X}$, or equivalently,

$$d^*(X) \in \text{sgn}(\inf_{s \in \mathbb{S}}\{E(Y|X,S=s,A=1) - \inf_{s \in \mathbb{S}}\{E(Y|X,S=s,A=-1)\}).$$

For a continuous $S$, we consider $G_{S|X}\{E(Y|X,S,A=d(X))\}$ as $Q^\tau_{S|X}\{E(Y|X,S,A=d(X))\}$, which is the $\tau$-th quantile of $E(Y|X,S,A=d(X))$ and $\tau \in (0,1)$ is the quantile level of interest. Specifically, $Q^\tau_{S|X}\{E(Y|X,S,A=d(X))\} = \inf\{t : F(t) \geq \tau\}$ with $F$ denoting the conditional distribution function of $E(Y|X,S,A=d(X))$ over $\mathbb{S}$ given $X$ and $d$. Note the randomness behind $E(Y|X,S,A=d(X))$ given $X$ and $d$ is fully determined by the sensitive variable $S$. Then optimal IDR under this criterion is defined as

$$d^* \in \arg\max_{\mathbb{D}} E_X\big[Q^\tau_{S|X}\{E(Y|X,S,A=d(X))\}\big].$$

This implies that for a given $X$, $d^*(X)$ assigns a treatment that yields the largest $\tau$-th quantile of the outcome over the distribution related to $S$, or equivalently,

$$d^*(X) \in \text{sgn}(\{Q^\tau_{S|X}\{E(Y|X,S,A=1)\} - Q^\tau_{S|X}\{E(Y|X,S,A=-1)\}).$$

We let $\tau = 0.25$ throughout the paper and suppress $\tau$ for simplicity. Results on varying the value of $\tau$ is provided in Appendix D; see Section 4.1 for details.

**Identifying vulnerable subjects.** Our RISE framework provides a natural way to define *vulnerable groups*. Specifically, for a discrete $S$, if $\inf_S\{E(Y|X,S,A=1)\} > \inf_S\{E(Y|X,S,A=0)\}$, then $\arg\inf_S\{E(Y|X,S,A=0)\}$ is vulnerable given $X$, otherwise $\arg\inf_S\{E(Y|X,S,A=1)\}$ is vulnerable. In other words, the vulnerable subjects are those in the worst-off group that needs protection. Similarly, for a continuous $S$, if $Q_S\{E(Y|X,S,A=1)\} > Q_S\{E(Y|X,S,A=0)\}$, then the set $\{S : E(Y|X,S,A=0) \leq Q_S\{E(Y|X,S,A=0)\}\}$ defines the vulnerable subjects given $X$, otherwise this group is defined as $\{S : E(Y|X,S,A=1) \leq Q_S\{E(Y|X,S,A=1)\}\}$.

### 3.3 Estimation and Algorithm

Here we provide a transformation of the proposed RISE from an optimization problem to a weighted classification problem. There are several advantages to this conversion: 1) The optimization problem defined in (2) involves a nonsmooth and nonconvex objective function that could lead to computational challenges. 2) With multiple powerful statistical and machine learning toolbox to choose from, a classification problem can be more readily solved in practice. Hyperparameter tuning and model selection could be conducted to further boost performance. 3) Compared to a direct optimization of (2), a classification-based optimizer allows the use of off-the-shelf software packages that can be tailored to different functional classes or incorporate different properties such as model sparsity.

**Proposition 1.** *Maximizing the objective function in* (2) *is equivalent to maximizing*

$$E_X\big\{\mathbb{1}(d(X)=1)[G_{S|X}\{E(Y|X,S,A=1)\} - G_{S|X}\{E(Y|X,S,A=-1)\}]\big\}.$$

With Proposition 1 and a proper estimator of the outcome model $E(Y|X,S,A)$ using training data $\mathcal{D}_n = \{X_i, S_i, A_i, Y_i\}_{i=1}^n$, we replace the expectation of $Y_i$ by its estimate $\hat{Y}_i$ and solve the following,

$$\arg\max_{d\in\mathbb{D}} n^{-1}\sum_{i=1}^n [\mathbb{1}(d(x_i)=1)\{g_1(x_i) - g_2(x_i)\}], \tag{3}$$

where $g_1(x_i) = G_{s|x}\{\hat{Y}_i(x_i, s, a_i = 1)\}$ and $g_2(x_i) = G_{s|x}\{\hat{Y}_i(x_i, s, a_i = -1)\}$. We have the following proposition to address noncontinuity in (3) and transform it into a classification problem. Define $\mathbb{F}$ as a class of all measurable functions over $\mathbb{X}$.

**Proposition 2.** *Maximizing the empirical objective in* (3) *is equivalent to a weighted classification problem of minimizing over* $f \in \mathbb{F}$,

$$n^{-1}\sum_{i=1}^n \mathbb{1}[\text{sgn}\{g_1(x_i) - g_2(x_i)\}\cdot f(x_i) < 0]\cdot|g_1(x_i) - g_2(x_i)| \tag{4}$$

*with features* $x_i$, *a label* $\text{sgn}\{g_1(x_i) - g_2(x_i)\}$ *and the sample weight* $|g_1(x_i) - g_2(x_i)|$ *for* $1 \le i \le n$.

With Proposition 2, we have transformed the optimization problem (2) into a weighted classification problem (4) where for subject $i$ with features $x_i$, the true label is $\text{sgn}\{g_1(x_i) - g_2(x_i)\}$ and the sample weight is $|g_1(x_i) - g_2(x_i)|$. The estimated optimal decision rule by (4) is then given by $\hat{d}(x) = \text{sgn}\{\hat{f}(x)\}$[3]. The proof of Proposition 1 and Proposition 2 is presented in Appendix B.

Algorithm 1 provides an algorithmic overview of RISE. The inner expectation $E(Y|X,S,A)$ can be modeled as $\hat{Y}(X,S,A)$ using a twin model separated by the treatment and control groups (i.e., a T-learner as in [27]). For a continuous $S$, $G(X,A) = Q_{S|X,A}\{E(Y|X,S,A)\}$ is estimated via a quantile regression of $\hat{Y}$ on $X$ but without $S$. For a discrete $S$, an estimate of $G(X,A) = \inf_S\{E(Y|X,S,A)\}$ is obtained by finding the minimum among $\{E(Y|X,S=1,A),\dots,E(Y|X,S=K,A)\}$. The estimated decision rule can then be obtained from the weighted classification. In our implementation, neural networks are used to fit models in the training data sets. The details on modeling and hyperparameter tuning via cross-validations are given in Appendix C. A Python package `rise` based on neural networks is available on GitHub (`https://github.com/ellenxtan/rise`). Note that the model choices are flexible.

---

**Algorithm 1:** RISE (Robust individualized decision learning with sensitive variables)

**Input:** Training data $\mathcal{D}_n = \{Y_i, A_i, X_i, S_i\}_{i=1}^n$
**Output:** Estimated decision rule $\hat{d}$
**for** $i \leftarrow 1$ **to** $n$ **do**
  $\hat{Y}_i(x_i, s_i, a_i) \leftarrow$ Model $E(Y|X,S,A=a)$ using $\mathcal{D}_n$ with $a = 1$ and $a = -1$, respectively;
  **if** *$S$ is continuous* **then**
    $g_1(x_i) \leftarrow$ Model $Q_{S|X,A}\{E(Y|X,S,A=a)\}$ via quantile regressions of $\hat{Y}_i(x_i, s_i, a_i)$ on $x_i$, for $\mathcal{D}_n$ with $a = 1$;
    $g_2(x_i) \leftarrow$ Model $Q_{S|X,A}\{E(Y|X,S,A=a)\}$ via quantile regressions of $\hat{Y}_i(x_i, s_i, a_i)$ on $x_i$, for $\mathcal{D}_n$ with $a = -1$;
  **else if** *$S$ is discrete* **then**
    $g_1(x_i) \leftarrow$ Compute $\inf_{s\in\mathbb{S}}\{\hat{Y}_i(x_i, s, a_i = 1)\}$;
    $g_2(x_i) \leftarrow$ Compute $\inf_{s\in\mathbb{S}}\{\hat{Y}_i(x_i, s, a_i = -1)\}$;
**end**
$\hat{d} \leftarrow$ Build a weighted classification model with features $x_i$, label $\text{sgn}\{g_1(x_i) - g_2(x_i)\}$, and sample weight $|g_1(x_i) - g_2(x_i)|$ for $1 \le i \le n$;
**return** $\hat{d}$.

---

[3]If $\hat{f}(x) = 0$, assign a random treatment.

## 3.4 Extension to Multiple Sensitive Variables

The extension from $S$ being a single continuous variable to multiple continuous variables is straightforward in Algorithm 1. For multiple discrete sensitive variables, similar estimation procedure can be conducted as outlined in Section 3.3. Suppose there are $L$ discrete sensitive variables, i.e., $\mathcal{S} = \{S_1, S_2, \ldots, S_L\}$. The inner expectation $E(Y|X, S_1, \ldots, S_L, A)$ can be obtained with a twin model of $Y$ on $X$ and all $\mathcal{S}$ for each treatment level. The infimum over $\mathbb{S}$ is obtained by finding the minimum iterating space of possible parameter values for each sensitive variable. See Section 4.2 for an example of using multiple discrete sensitive variables. We will discuss in Section 5 the challenges and future work related to the scenario with a mixture of continuous and discrete sensitive variables and the identification of vulnerable subjects under these cases.

# 4 Numerical Studies

In this section, we perform extensive numerical experiments to investigate the merit of robustness of the proposed framework via simulations and three real-data applications. The results demonstrate that the proposed rules achieve a robust objective with sensitive variables unavailable at the time of decision while maintaining comparable mean outcomes.

**Compared approaches.** For comparison, we consider the naive and mean-optimal approaches described in Section 3.1, which correspond to different choices of $G(\cdot)$ functions. The naive decision rule that simply disregard information of $S$, denoted as **Base**, can be formulated in our optimization framework of (2) by letting $G(X, A) = E(Y|X, A)$. The IDR can be estimated directly by fitting a model of $Y$ on $X$ in each treatment arm. The resulting IDR is not sensitive variables-aware and is biased due to confounding, as discussed. Another IDR that resembles traditional mean-optimal decision rules, denoted as **Exp**, can be formulated as $G(X, S, A) = E(Y|X, S, A)$. This can be obtained by training a classification model without $S$, i.e., only using $X$, after obtaining an outcome model for the inner expectation $E(Y|X, S, A)$. Note that this approach is not robust to extreme behaviors in $S$. The modeling approaches described in Appendix C apply to here. We also include the *double robust* [11, 68] versions of Base and Exp, respectively, by adapting Policytree (PT) [64, 3], the latest state-of-the-art policy learning method for maximizing the expected values. The double robust analogues of Base and Exp are termed **PT-Base** and **PT-Exp**, respectively.

**Evaluation metrics.** *1) Objective:* the quantile objective is estimated and reported for a continuous $S$ and the infimum objective is for a discrete $S$. The objective, when $\tau < 0.5$, (here $\tau = 0.25$) represents the value of the "low performers" among all possible value of $S$ under a given $d$. *2) Value:* the value function, or expected reward used by the most existing methods, such as [36, 52], is defined as $V(d) = E\{Y(d)\}$. It represents the "average performers". For randomized trials, an unbiased estimator of $V(d)$ is given by $\hat{V}(d) = \{\sum_{i=1}^{T} Y_i \mathbb{1}(A_i = \hat{d}(X_i))/\pi(A_i, X_i)\}/\{\sum_{i=1}^{T} \mathbb{1}(A_i = \hat{d}(X_i))/\pi(A_i, X_i)\}$ [41], where $T$ is the sample size of the test data and $\pi(A, X)$ is propensity score. For observational studies, the value is estimated with $\hat{V}(d) = T^{-1}\sum_{i=1}^{T} \hat{Y}_i(x_i, s_i, a_i = \hat{d})$. We report the metrics among all subjects and among the potential vulnerable subgroup, respectively. For simulation, we consider training data and testing data with sample sizes of 8,000 and 2,000, respectively. For real-data applications, we consider a 80-20 split of the dataset into a training data and a testing data. Continuous covariates are standardized before the estimation. All results are based on 100 replications. Experiments are performed on a 6-core Intel Xeon CPU E5-2620 v3 2.40GHz equipped with 64GB RAM.

## 4.1 Simulation Studies

*Example 1.* Here we provide details for the simulation of the motivating example introduced in Section 1. The outcome is generated according to: $Y_i = \mathbb{1}(X_i > 0.5)\{5 + 10\mathbb{1}(A_i = 1) + 22S_i - 24\mathbb{1}(A_i = 1)S_i\} + \mathbb{1}(X_i \le 0.5)\{11 + 19\mathbb{1}(A_i = 1) + 2S_i - 32\mathbb{1}(A_i = 1)S_i\} + \epsilon_i$, where the covariate $X_i \sim Unif(0, 1)$, treatment assignment $A_i \sim Bernoulli(0.5)$, and the noise $\epsilon_i \sim N(0, 1)$. For a discrete type $S$, $S_i \sim Bernoulli(0.5)$. For a continuous type $S$, $S_i$ is generated from a mixture of beta distributions, $Beta(4, 1)$ and $Beta(1, 4)$, with equal mixing proportions.

*Example 2.* We generate $Y$ using the following model: $Y_i = \{0.5 + \mathbb{1}(A_i = 1) + \exp(S_i) - 2.5S_i\mathbb{1}(A_i = 1)\}\{1 + X_{i1} - X_{i2} + X_{i3}^2 + \exp(X_{i4})\} + \{1 + 2\mathbb{1}(A_i = 1) + 0.2\exp(S_i) - 3.5S_i\mathbb{1}(A_i = $

1)$\}\{1+5X_{i1}-2X_{i2}+3X_{i3}+2\exp(X_{i4})\}+\epsilon$, where $X_{ij}\sim Unif(0,1)$, $j=1,\ldots,6$, $A$ satisfies $\log\{P(A_i=1|X_i)/P(A=0|X_i)\}=0.6(-S_i+X_{i1}-X_{i2}+X_{i3}-X_{i4}+X_{i5}-X_{i6})$, and $\epsilon_i\sim N(0,1)$. For a continuous type $S$, $S_i$ is generated from a mixture of beta distributions, $Beta(4,1)$ and $Beta(1,4)$, with equal mixing proportions; for a discrete type $S$, we consider a binary $S_i$ that satisfies $\log\{P(S_i=1|X_i)/P(S_i=0|X_i)\}=-2.5+0.8(X_{i1}+X_{i2}+X_{i3}+X_{i4}+X_{i5}+X_{i6})$.

Table 3 summarizes the performance of the proposed IDRs compared to the mean criterion for Example 1 and Example 2. The proposed RISE achieves the largest objectives and improves the value among vulnerable subjects, while maintaining comparative overall values. As for the objective, intuitively, the proposed rule is expected to achieve a larger objective than all other methods uniformly in $\mathbb{X}$. We also point out that there is no direct relationship between the objective among all subjects versus the objective among vulnerable subjects. For example, using the toy example with setup in Table 1, and limiting to subjects with $X\le0.5$ only, $S=1$ is vulnerable and is assigned $A=-1$ by the proposed RISE. The objective among $S=1$ is 13 but the objective among both $S=0$ and $S=1$ is $12=(11+13)/2$, which is smaller than that among the vulnerable group. In other words, by protecting the vulnerable subjects, the proposed rule may lead to an increase in the outcome of the vulnerable group, and the gain may result in a higher outcome than the overall mean outcome. PT-Exp tends to show the best improvement in terms of the overall value, as the doubly robust-based estimators tend to reduce variance in value estimation. However, PT-Exp is shown to have minimal benefits for vulnerable subjects. RISE still shows the largest gain in the objective and value among vulnerable subjects among all compared methods.

Table 3: Simulation results for Example 1 and Example 2. Standard error in parenthesis.

| Example | Type of $S$ | IDR | Obj. (all) | Obj. (vulnerable) | Value (all) | Value (vulnerable) |
|---|---|---|---|---|---|---|
| 1 | Disc. | Base | 7.03 (0.03) | 7.01 (0.04) | 14.3 (0.05) | 7.92 (0.06) |
| | | Exp | 6.39 (0.03) | 6.39 (0.04) | 14.4 (0.05) | 7.14 (0.06) |
| | | PT-Base | 2.66 (0.02) | 2.65 (0.02) | 15.4 (0.05) | 2.58 (0.02) |
| | | PT-Exp | 2.62 (0.02) | 2.62 (0.02) | **15.5** (0.05) | 2.55 (0.02) |
| | | RISE | **12.0** (0.01) | **12.0** (0.01) | 13.0 (0.01) | **14.0** (0.01) |
| | Cont. | Base | 9.12 (0.03) | 9.14 (0.04) | 14.5 (0.08) | 8.25 (0.11) |
| | | Exp | 8.75 (0.03) | 8.75 (0.04) | 14.6 (0.08) | 7.58 (0.06) |
| | | PT-Base | 6.71 (0.03) | 6.72 (0.03) | 15.3 (0.05) | 4.52 (0.02) |
| | | PT-Exp | 6.68 (0.02) | 6.67 (0.02) | **15.4** (0.05) | 4.47 (0.02) |
| | | RISE | **12.2** (0.02) | **12.2** (0.03) | 13.0 (0.01) | **13.7** (0.01) |
| 2 | Disc. | Base | 7.79 (0.02) | 8.66 (0.03) | 19.4 (0.04) | 11.4 (0.06) |
| | | Exp | 9.12 (0.03) | 10.1 (0.03) | **19.5** (0.04) | 14.4 (0.05) |
| | | PT-Base | 7.19 (0.03) | 7.77 (0.03) | 19.0 (0.05) | 9.71 (0.05) |
| | | PT-Exp | 8.30 (0.02) | 9.03 (0.03) | 19.1 (0.04) | 12.2 (0.05) |
| | | RISE | **13.5** (0.01) | **14.0** (0.01) | 17.4 (0.02) | **22.1** (0.02) |
| | Cont. | Base | 9.89 (0.02) | 9.87 (0.03) | 17.6 (0.02) | 9.09 (0.04) |
| | | Exp | 11.1 (0.02) | 11.1 (0.02) | 17.8 (0.02) | 12.2 (0.04) |
| | | PT-Base | 9.30 (0.02) | 9.29 (0.03) | 18.0 (0.03) | 7.61 (0.04) |
| | | PT-Exp | 9.41 (0.02) | 9.41 (0.02) | **18.1** (0.02) | 7.92 (0.04) |
| | | RISE | **14.1** (0.01) | **14.2** (0.02) | 17.0 (0.01) | **20.3** (0.03) |

In the appendix, we consider a continuous $S$ for different quantile criteria $\tau=0.1$ and $0.5$ to test the robustness of RISE. Results show that when $\tau$ is small, there is more strength in the proposed method, as the algorithm aims to improve the worst-outcome scenarios. The proposed RISE has the largest gain in objective and value among vulnerable subjects when $\tau$ is $0.1$, and has similar performance as the compared approaches when $\tau$ is $0.5$. We also consider a scenario where $S$ is not involved in the data generation of $Y$, i.e., Assumption 1c is simplified as $\{Y(-1),Y(1)\}\perp A|X$. The estimated objective and value function are similar across all compared approaches, which indicates the robustness of RISE. Finally, we study the performances of our method when Assumption 1b is nearly violated or Assumption 1c is violated. Similar patterns have been observed that the proposed RISE achieves the largest objectives and improves the value among vulnerable subjects, while maintaining comparable overall values. The details can be found in Appendix D.

## 4.2 Real-data Applications

We present three real-data examples to showcase the robust performance of RISE. These applications consider either fairness or safety in the context of policy [30] and healthcare [19, 59] where sensitive variables commonly exist.

**Fairness in a job training program.** To illustrate the implication of the proposed method from a fairness perspective, we consider the National Supported Work (NSW) program [30] for improving personalized recommendations of a job training program on increasing incomes. This program intended to provide a 6 to 18-month training for individuals in face of economical and social stress such as former drug addicts and juvenile delinquents. The original experimental dataset consists of 185 individuals who received the job training program ($A = 1$) and 260 individuals who did not ($A = -1$). The baseline covariates are age, years of schooling, race (1 = African Americans or Hispanics, 0 = others), married (1 = yes, 0 = no), high school diploma (1 = yes, 0 = no), earning in 1974, and earning in 1975. The outcome variable is the earning in 1978. In the exploratory analysis using causal forest [71], a random forest-based method for causal inference, we observe that age may play an important role in the causal effect of the job training program on the long-term post-market earning. In this application example we use age as the sensitive variable $S$ and other baseline covariates as $X$. The earnings in years 1974, 1975, and 1978 are transformed by taking the logarithm of the earning plus one.

**Improvement of HIV treatment.** To illustrate the implication of the proposed method from a safety perspective when there is delayed information, we consider the ACTG175 dataset among HIV positive patients [19]. The original study considers a total of 2,139 patients who were randomly assigned into four treatment groups. In this data application, we focus on finding the optimal IDRs between two treatments: zidovudine combined with didanosine ($A = -1$) and zidovudine combined with zalcitabine ($A = 1$). The total number of patients receiving these two treatments is 1,046. The baseline covariates we consider are age, weight, CD4 T-cell amount at baseline, hemophilia (1 = yes, 0 = no), homosexual activity (1 = yes, 0 = no), Karnofsky score, history of intravenous drug use (1 = yes, 0 = no), gender (1 = male, 0 = female), CD8 T-cell amount at baseline, race (1 = non-Caucasian, 0 = Caucasian), number of days of previously received antiretroviral therapy, use of zidovudine in the 30 days prior to treatment initiation (1 = yes, 0 = no), and symptomatic indicator (1 = symptomatic, 0 = asymptomatic). The outcome variable is the CD4 T-cell amount at $96 \pm 5$ weeks from the baseline. We consider CD8 T-cell amount at baseline as the sensitive variable. The response of CD8 T-cell among HIV positive patients has not been fully understood [7]. Clinically, it is plausible that only CD4 is measured in clinical visits where treatments are based on, hence CD8 might not be measured and not used in decision making. As our exploratory analysis using causal forest shows, CD8 T-cell amount may play an important part in the treatment effect of the outcome.

**Safe resuscitation for patients with sepsis.** For this application, we apply the proposed method to treating sepsis, a life-threatening disease. This application intends to provide an example to apply our method with multiple categorical sensitive variables in the scenario where there is missing yet important information at the time of decision making. We apply the proposed method to a sepsis study from the University of Pittsburgh Medical Center (UPMC). The original study cohort includes 30,687 patients with Sepsis-3 [59] within 6 hours of hospital arrival from 14 UPMC hospitals between 2013 and 2017. For our data analysis, we consider $X$ to be baseline patient characteristics 4 hours before sepsis onset, which includes patient demographics of age, gender (1 = male, 0 = female), race (1 = Caucasian, 0 = others), and weight, and vital signs of usage of mechanical ventilation (1 = yes, 0 = no), respiratory rate, temperature, intravenous fluids (1 = yes, 0 = no), Glasgow Coma Scale score, platelets, blood urea nitrogen, white blood cell counts, glucose, creatinine. We consider two sensitive variables, lactate and Sequential Organ Failure Assessment (SOFA) score 4 hours before sepsis onset. Lactate and SOFA score have been two important indicators of sepsis severity [21, 26, 60, 62]. Different from the baseline patient demographics or common vital signs that are typically obtained at the admission of patients, SOFA score combines performance of several organ systems in the body [59], which requires additional calculation and cannot be obtained directly. Lactate labs measures the level of lactic acid in the blood [2] and are less common in routine examination, which could be delayed in ordering. Hence, their measurements are obtained retrospectively after treatment decisions have been made and are not available at times of decision. We dichotomize lactate level at clinically meaningful value of 2 mmol/L [60], and SOFA score at value of 6 for analysis [70, 18]. The treatment option is whether the patient took any vasopressors during the first 24 hours after sepsis onset. The outcome is hospital survival ($Y = 1$) or death ($Y = 0$). The analysis cohort contains 6,539 patients

in total. We are interested in making decision about whether to treat patients with vasopressors in the first 24 hours after sepsis onset given the measurements of lactate and SOFA are not available at the time of decision making. Additional rationale and background on this example are provided in Appendix E.2.

Table 4: Estimated objective and value of different IDRs for the three data applications. Standard error in parenthesis. The outcome of each study is italicized.

| Dataset | IDR | Obj. (all) | Obj. (vulnerable) | Value (all) | Value (vulnerable) |
|---|---|---|---|---|---|
| NSW *log(income+1)* | Base | 5.26 (0.04) | 5.28 (0.05) | 6.32 (0.05) | 6.33 (0.07) |
| | Exp | 5.22 (0.04) | 5.24 (0.05) | 6.37 (0.05) | 6.37 (0.07) |
| | PT-Base | 4.97 (0.04) | 5.08 (0.06) | 6.40 (0.03) | 6.38 (0.05) |
| | PT-Exp | 5.03 (0.04) | 5.11 (0.05) | **6.43** (0.03) | 6.40 (0.05) |
| | RISE | **5.43** (0.04) | **5.44** (0.04) | 6.42 (0.04) | **6.42** (0.06) |
| ACTG175 *CD4 T-cell amount* | Base | 336.9 (1.65) | 338.1 (2.23) | 353.5 (1.86) | 357.5 (2.24) |
| | Exp | 337.5 (1.65) | 338.9 (1.80) | 355.9 (1.95) | 359.1 (2.21) |
| | PT-Base | 299.7 (1.01) | 299.5 (1.91) | 356.9 (1.72) | 350.7 (2.54) |
| | PT-Exp | 300.1 (0.99) | 299.9 (1.83) | **357.1** (1.55) | 352.7 (2.61) |
| | RISE | **351.5** (1.67) | **351.2** (1.80) | 351.8 (1.88) | **363.1** (2.19) |
| Sepsis *survival rate* | Base | 0.803 (0.001) | 0.822 (0.001) | 0.965 (0.001) | 0.905 (0.002) |
| | Exp | 0.803 (0.001) | 0.822 (0.002) | 0.966 (0.001) | 0.908 (0.002) |
| | PT-Base | 0.758 (0.001) | 0.771 (0.002) | 0.981 (0.001) | 0.848 (0.003) |
| | PT-Exp | 0.758 (0.001) | 0.772 (0.002) | **0.984** (0.001) | 0.875 (0.003) |
| | RISE | **0.836** (0.001) | **0.833** (0.001) | 0.972 (0.001) | **0.923** (0.002) |

**Results.** Table 4 presents the performance of various IDRs on the three applications. As expected, RISE has the largest objective as well as value among vulnerable subjects. The patterns are similar to that in Section 4.1. We apply the Shapley additive explanations (SHAP) approach [33] to help visualize and interpret the important covariates in the decision rules given by RISE and Exp, respectively, in Appendix E.3. The SHAP approach provides unified values to describe the correlation between each feature and the predicted decision rule, respectively [33]. Overall, the direction of correlations is similar for RISE and Exp, but their ranking of feature importance may be different.

## 5  Discussion

We have proposed RISE, a robust decision learning framework with a novel quantile- or infimum-optimal treatment objective intended to improve the worst-case scenarios of individuals when decisions with uncertainty need to be made, but with sensitive yet important information missing. Our approach can be applied to a broad range of applications, including but not limited to policy, education, healthcare, etc. For a mixture of continuous and discrete sensitive variables, the estimated rule can be obtained by first taking the infimum over the discrete ones as in Section 3.4, then obtaining the quantile over the continuous ones. However, challenges remain in finding the vulnerable subjects described in Section 3.2 under these settings as it may be computationally difficult to find a vulnerable set of $S$ when it is multi-dimensional. Another future work includes the extension of the current binary treatment option to a multi-treatment option. It is also worth mentioning that our work can be naturally extended to the scenario where there exist unmeasured confounders. As long as the conditional average outcome given observed covariates can be identified (via instrumental variables such as [73] or negative control variables such as [50]), our method can be applied.

## Acknowledgements

This research was supported in part by the Competitive Medical Research Fund of the UPMC Health System and the University of Pittsburgh Center for Research Computing through the resources provided. The authors thank Jason N. Kennedy for the sepsis data preparation, and Emily B. Brant, Chung-Chou H. Chang and Gong Tang for helpful discussions and feedback.

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
