## A  Additional Literature Review

Typical model-based methods for deriving IDRs include Q-learning such as [75, 39, 38] and A-learning such as [53, 40] where a model of responses is imposed and the optimal decision rule is obtained by optimizing value function derived from the model. Model-based methods posit a model of responses given observed covariates and treatment assignments, and obtain the optimal IDR by optimizing the corresponding value function derived from the model. Q-learning optimizes the corresponding value function derived from a parametric model of responses given observed covariates and treatment assignments, and it results in an optimal decision rule. A-learning is a semiparametric method, which derives from a model that directly describes the difference between treatments, with the baseline remaining unspecified. On the other hand, model-free methods such as [77, 81, 80] assign values to actions simply through trial and error without pre-specifying a model. Besides, contextual bandit methods (see [6] and references therein) test out different actions and automatically learn which one has the most rewarding outcome for a given situation. See [9, 8, 28, 24] and references therein for a comprehensive review on general IDRs under causal settings.

## B  Proof of Propositions

*Proof of Proposition 1.* We observe that to maximize the objective function in (2) is equivalent to maximizing

$$
\begin{aligned}
&E_X\big[G_{S|X}\{E(Y|X,S,A=d(X))\}|X\big] \\
&= E_X\big[G_{S|X}\{E(Y|X,S,A=1)\}\mathbb{1}(d(X)=1) \\
&\qquad + G_{S|X}\{E(Y|X,S,A=-1)\mathbb{1}(d(X)=-1)\}\big] \\
&= E_X\big\{\mathbb{1}(d(X)=1)[G_{S|X}\{E(Y|X,S,A=1)\} - G_{S|X}\{E(Y|X,S,A=-1)\}] \\
&\qquad + G_{S|X}\{E(Y|X,S,A=-1)\}\big\} \\
&\propto E_X\big\{\mathbb{1}(d(X)=1)[G_{S|X}\{E(Y|X,S,A=1)\} - G_{S|X}\{E(Y|X,S,A=-1)\}]\big\}.
\end{aligned}
$$

$\square$

*Proof of Proposition 2.* Let $d(x) = \mathrm{sgn}\{f(x)\}$, by this transformation, we consider the following objective on a smooth function $f(x)$,

$$
\begin{aligned}
&\arg\max_{d\in\mathbb{D}}\tfrac{1}{n}\sum_{i=1}^{n}\big\{\mathbb{1}(d(x_i)=1)[g_1(x_i)-g_2(x_i)]\big\} \\
&= \arg\max_f \tfrac{1}{n}\sum_{i=1}^{n}\mathbb{1}[\mathrm{sgn}\{f(x_i)\}=1]\cdot[g_1(x_i)-g_2(x_i)] \\
&= \arg\min_f \tfrac{1}{n}\sum_{i=1}^{n}\mathbb{1}\{1\cdot f(x_i)<0\}\cdot[g_1(x_i)-g_2(x_i)] \\
&= \arg\min_f \tfrac{1}{n}\sum_{i=1}^{n}\mathbb{1}[\mathrm{sgn}\{g_1(x_i)-g_2(x_i)\}\cdot f(x_i)<0]\cdot|g_1(x_i)-g_2(x_i)|.
\end{aligned}
$$

The sign of the estimated $f$ above is a solution of $d$ to (3).

Hence, the proposed classification-based objective is to minimize

$$
\tfrac{1}{n}\sum_{i=1}^{n}\mathbb{1}[\mathrm{sgn}\{g_1(x_i)-g_2(x_i)\}\cdot f(x_i)<0]\cdot|g_1(x_i)-g_2(x_i)|.
$$

To this point, we have transformed the optimization problem (2) into a weighted classification problem where for subject $i$ with features $x_i$, the true label is $\mathrm{sgn}\{g_1(x_i)-g_2(x_i)\}$ and the sample weight is $|g_1(x_i)-g_2(x_i)|$. $\square$

## C  Details on Modeling and Hyperparameter Tuning

In our implementation, neural networks with mean or quantile losses are used to fit the models with hyperparameters tuned via a 5-fold cross validation in the training data sets. Specifically, implemented in TensorFlow [1], neural networks with mean squared loss is used to model $E(Y|X,S,A)$ separated by the control arm and the treatment arm, respectively. For continuous $S$, to model $Q_{S|X,A}\{E(Y|X,S,A)\}$, neural networks with quantile loss is used with a prespecified $\tau$, for the control arm and the treatment arm, respectively. In the final weighted classification model, neural

networks with cross-entropy loss is used. Note that the model choices here are flexible. One can perform model selection if they would like to.

Hyperparameter tuning helps prevent overfitting and is essential in machine learning methods or other black-box algorithms such as neural networks. In our implementation, the optimal hyperparameters are obtained via a 5-fold cross validation in the training data sets. Specifically, we consider the number of hidden layers (1, 2, and 3 layers), the number of hidden units in each layer (256, 512, and 1024 nodes), activation function (RELU, Sigmoid, and Tanh), optimizer (Adam, Nadam, and Adadelta), dropout rate (0.1, 0.2, and 0.3), number of epochs (50, 100, and 200), and batch size (32, 64, and 128).

# D  Additional Simulations

**Different quantile criteria.**  For the quantile criteria, we also consider $\tau = 0.1$ and $0.5$, respectively. Table 5 presents the simulation results for Example 2 with continuous $S$ using 0.1 quantile criterion and 0.5 quantile criterion, respectively. Results show that when $\tau$ is small, there is more strength in the proposed method, as the algorithm aims to improve the worst-outcome scenarios. The proposed RISE has the largest gain in objective and value among vulnerable subjects when $\tau$ is 0.1, and has similar performance as the compared approaches when $\tau$ is 0.5.

$S$ **as a noise variable.**  We generate the outcome $Y$ using the following model where $S$ is not involved: $Y = \mathbb{1}(X_1 \leq 0.5)\{8 + 12\mathbb{1}(A = 1) + 16\exp(X_2) - 26\mathbb{1}(A = 1)X_2\} + \mathbb{1}(X_1 > 0.5)\{13 + 3\mathbb{1}(A_i = 1) + 2\exp(X_2) - 8\mathbb{1}(A = 1)X_2\} + \epsilon$, where $X_j \sim Unif(0, 1)$, $j = 1, 2$, $A \sim Bernoulli(0.5)$, and $\epsilon \sim N(0, 1)$. For continuous $S$, $S = \text{expit}\{-2.5(1 - X_1 - X_2)\}$; for discrete $S$, we consider a binary $S$ that satisfies $\log\{P(S = 1|X)/P(S = 0|X)\} = -2.5(1 - X_1 - X_2)$. Table 6 summarizes the performance of the proposed IDRs compared to the mean criterion for Example 2. The estimated objective and value function are similar for the compared IDRs, which indicates the robustness of the proposed RISE.

**Violations of causal assumptions.**  To further test the robustness of the proposed RISE, we consider scenarios where Assumption 1 may not hold. To test the violation of positivity assumption, using the same setting as in Example 2, we consider an extreme propensity score, or the probability of being treated given $X$ and $S$. Specifically, we let $A$ satisfy $\log\{P(A_i = 1|X_i)/P(A = 0|X_i)\} = -1.2(-S_i + X_{i1} - X_{i2} + X_{i3} - X_{i4} + X_{i5} - X_{i6})$. To test the unconfoundedness assumption, a random normal noise, $e \sim N(0, 1)$ is added to $X_1$ in the setting of Example 2. The simulation results are presented in Table 7 and Table 8 respectively.

Table 5: Simulation results for Example 2 with continuous $S$ using 0.1 quantile criterion and 0.5 quantile criterion, respectively. Standard error in parenthesis. The proposed RISE has more strengths when $\tau$ is small, as the algorithm aims to improve the worst-outcome scenarios.

| Type of $S$ | $\tau$ | IDR | Obj. (all) | Obj. (vulnerable) | Value (all) | Value (vulnerable) |
|---|---|---|---|---|---|---|
| Cont. | 0.1 | Base | 7.93 (0.03) | 7.92 (0.03) | 17.7 (0.02) | 8.64 (0.07) |
| | | Exp | 8.88 (0.05) | 8.85 (0.05) | 17.8 (0.02) | 10.6 (0.12) |
| | | PT-Base | 6.97 (0.02) | 6.95 (0.02) | 17.9 (0.03) | 6.65 (0.04) |
| | | PT-Exp | 7.11 (0.02) | 7.08 (0.03) | **18.0** (0.03) | 6.96 (0.05) |
| | | RISE | **13.8** (0.01) | **13.7** (0.02) | 16.9 (0.01) | **20.9** (0.03) |
| Cont. | 0.5 | Base | 17.3 (0.04) | 17.2 (0.04) | 17.7 (0.02) | 23.8 (0.19) |
| | | Exp | 17.2 (0.03) | **17.4** (0.03) | 17.8 (0.02) | 22.1 (0.17) |
| | | PT-Base | 17.3 (0.05) | 17.3 (0.05) | 18.0 (0.03) | 23.9 (0.26) |
| | | PT-Exp | **17.4** (0.05) | **17.4** (0.05) | **18.1** (0.03) | **24.0** (0.25) |
| | | RISE | **17.4** (0.04) | **17.4** (0.04) | 17.8 (0.02) | **24.0** (0.22) |

Table 6: Simulation results for scenario when $S$ is a noise variable. Vulnerable subjects cannot be defined as $S$ is not important in the example. The estimated objective and value function are similar for the compared IDRs, which indicates the robustness of the proposed RISE.

| Type of $S$ | IDR | Obj. (all) | Obj. (vulnerable) | Value (all) | Value (vulnerable) |
|---|---|---|---|---|---|
| | Base | 27.5 (0.03) | - | 27.5 (0.06) | - |
| | Exp | 27.5 (0.03) | - | 27.5 (0.06) | - |
| Disc. | PT-Base | 27.5 (0.02) | - | 27.5 (0.03) | - |
| | PT-Exp | 27.5 (0.02) | - | 27.5 (0.03) | - |
| | RISE | 27.5 (0.03) | - | 27.5 (0.06) | - |
| | Base | 27.2 (0.04) | - | 27.3 (0.07) | - |
| | Exp | 27.2 (0.04) | - | 27.3 (0.07) | - |
| Cont. | PT-Base | 27.2 (0.04) | - | 27.3 (0.07) | - |
| | PT-Exp | 27.2 (0.04) | - | 27.3 (0.07) | - |
| | RISE | 27.2 (0.04) | - | 27.3 (0.07) | - |

Table 7: Simulation results for Example 2 where the positivity assumption in Assumption 1b is nearly violated. Standard error in parenthesis.

| Type of $S$ | IDR | Obj. (all) | Obj. (vulnerable) | Value (all) | Value (vulnerable) |
|---|---|---|---|---|---|
| | Base | 10.0 (0.03) | 11.1 (0.03) | 19.3 (0.02) | 16.1 (0.04) |
| | Exp | 8.80 (0.03) | 9.77 (0.04) | **19.5** (0.02) | 13.6 (0.04) |
| Disc. | PT-Base | 9.88 (0.03) | 10.7 (0.04) | 18.9 (0.02) | 15.4 (0.05) |
| | PT-Exp | 8.42 (0.03) | 9.14 (0.04) | 19.1 (0.02) | 12.4 (0.05) |
| | RISE | **13.5** (0.01) | **14.0** (0.01) | 17.3 (0.01) | **22.0** (0.02) |
| | Base | 11.5 (0.03) | 11.5 (0.04) | 17.5 (0.03) | 13.1 (0.04) |
| | Exp | 10.4 (0.04) | 10.4 (0.05) | 17.8 (0.04) | 10.3 (0.05) |
| Cont. | PT-Base | 11.0 (0.04) | 10.9 (0.04) | 17.7 (0.02) | 11.8 (0.04) |
| | PT-Exp | 9.63 (0.03) | 9.61 (0.03) | **18.0** (0.02) | 8.38 (0.03) |
| | RISE | **14.3** (0.01) | **14.3** (0.02) | 16.9 (0.01) | **20.4** (0.02) |

Table 8: Simulation results for Example 2 where the unconfoundedness assumption in Assumption 1c is violated. Standard error in parenthesis.

| Type of $S$ | IDR | Obj. (all) | Obj. (vulnerable) | Value (all) | Value (vulnerable) |
|---|---|---|---|---|---|
| | Base | 7.65 (0.04) | 8.44 (0.05) | 19.3 (0.03) | 11.1 (0.06) |
| | Exp | 8.94 (0.05) | 9.91 (0.06) | **19.4** (0.02) | 13.9 (0.06) |
| Disc. | PT-Base | 6.84 (0.03) | 7.35 (0.04) | 18.9 (0.03) | 8.91 (0.05) |
| | PT-Exp | 7.95 (0.05) | 8.62 (0.06) | 19.1 (0.03) | 11.4 (0.06) |
| | RISE | **13.5** (0.01) | **14.0** (0.01) | 17.4 (0.01) | **22.1** (0.02) |
| | Base | 9.58 (0.03) | 9.58 (0.03) | 17.9 (0.02) | 8.33 (0.05) |
| | Exp | 10.2 (0.04) | 10.2 (0.04) | 17.8 (0.02) | 9.83 (0.06) |
| Cont. | PT-Base | 9.27 (0.02) | 9.26 (0.03) | 17.9 (0.02) | 7.51 (0.03) |
| | PT-Exp | 9.34 (0.02) | 9.34 (0.03) | **18.0** (0.02) | 7.72 (0.03) |
| | RISE | **14.2** (0.01) | **14.1** (0.02) | 16.9 (0.01) | **20.1** (0.03) |

# E    Additional Information and Results for Real-data Applications

## E.1    Data Availability

The job training dataset [30] is available at `https://users.nber.org/~rdehejia/data/.nswdata2.html`. The ACTG175 dataset [19] is available in the R package `speff2trial`. The sepsis dataset [59] is proprietary and not publicly available. All data used in this work are deidentified.

### E.2   Additional Background on the Sepsis Application

Sepsis is leading cause of acute hospital mortality and commonly results in multi-organ dysfunction among ICU patients [57, 45]. Clinically, treatment decisions for sepsis patients are needed to be made within a short period of time due to the rapid deterioration of patient conditions. Lactate and the Sequential Organ Failure Assessment (SOFA) score have been two important indicators of sepsis severity and has been found to be more useful for predicting the outcome of sepsis than other clinical vitals and comorbidity scores [21, 26, 60, 34]. Typically, information of baseline patient characteristics such as age, gender, race, and weight, and common vital signs such as usage of mechanical ventilation, respiratory rate, temperature, intravenous fluids, Glasgow Coma Scale score, platelets, blood urea nitrogen, white blood cell counts, glucose, and creatinine are obtained at the admission of patients. On the other hand, SOFA score combines performance of several organ systems in the body such as neurologic, blood, liver, and kidney [59, 32, 67, 25] and cannot be obtained directly. Lactate labs measures the level of lactic acid in the blood [2, 48, 14] and are less common in routine examination, which could be delayed in ordering. Hence, their information may not be available by the time of treatment decision due to multiple reasons including doctors' delayed ordering, long laboratory processing time, or the rapid deterioration of development of sepsis, which poses tremendous difficulties for early diagnosis and treatment decisions within a short time. According to the new definition of Sepsis-3 [60], a serum lactate level greater than 2 mmol/L is considered to be in critical conditions and is highly likely to indicate a septic shock. Also, a SOFA score greater than 6 has been associated with a higher mortality [70, 18].

### E.3   Visualizations

We provide visualizations of features that are important in the estimated decision rules for the three real-data applications in Section 4.2. The Shapely additive explanations (SHAP) [33] is considered to be a united approach to explaining the predictions of any machine learning or black-box models. Figure 2, Figure 3, and Figure 4 presents the SHAP variable importance plots in the final weighted classification model by RISE and Exp, respectively, for the three real-data applications. Correlations between the feature and their SHAP value are highlighted in color. The red color means a feature is positively correlated with assigning treatment A = 1 and the blue indicates a negative correlation. Overall, the direction of correlation is similar for RISE and Exp, but their ranking of feature importance may be different.

**Fairness in a job training program.**  Figure 2 presents the SHAP variable importance plots in the final weighted classification model by RISE and Exp, respectively. We observe that whether having a high school diploma and income in 1974 are two important features in the variable important plot by RISE, while incomes in 1974 and in 1975 are important by Exp. It seems that being no degree and low income in 1974 has a higher chance of assigning $A = 1$ (to receive the job training program) by RISE, while low income in 1974 and but a higher income in 1975 may be associated with assigning $A = 1$ by Exp.

**Improvement of HIV treatment.**  Figure 3 presents the SHAP variable importance plots in the final weighted classification model by RISE and Exp, respectively. We observe that age and CD4 T-cell counts are two important features in the variable important plot by RISE, while weight and number of days of previously received antiretroviral therapy are important by Exp. It seems that being of a younger age and high CD4 T-cell count has a higher chance of assigning $A = 1$ (zidovudine combined with didanosine) by RISE, while being of a larger weight and few days of previously received antiretroviral therapy may be associated with assigning the treatment by Exp.

**Safe resuscitation for patients with sepsis.**  Figure 4 presents the SHAP variable importance plots in the final weighted classification model by RISE and Exp, respectively. We observe that Glasgow Coma Scale score, age, and platelets appears to be important features in both the plot by RISE and that by Exp. Other important features in the plot by RISE include temperature and blood urea nitrogen, where in the plot by Exp, respiratory rate and white blood cell counts are of top importance. Being in a low temperature with a high blood urea nitrogen tends to be predicted as $A = 1$ (to assign vasopressors) by RISE while being of higher respiratory rate with high white blood cell counts tends to be predicted as $A = 1$ by Exp.

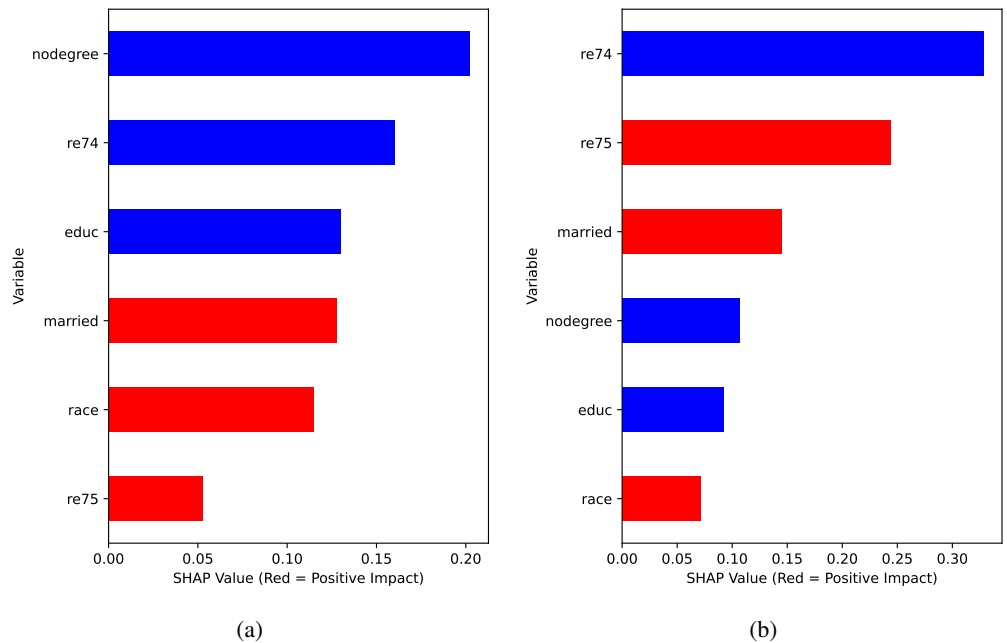

Figure 2: Visualization for the job training program: SHAP variable importance plots for decision rules RISE (a) and Exp (b), respectively. Covariates ($X$) are ranked by variable importance in descending order. Correlations between the feature and their SHAP value are highlighted in color. The red color means a feature is positively correlated with assigning treatment $A = 1$ and the blue indicates a negative correlation.

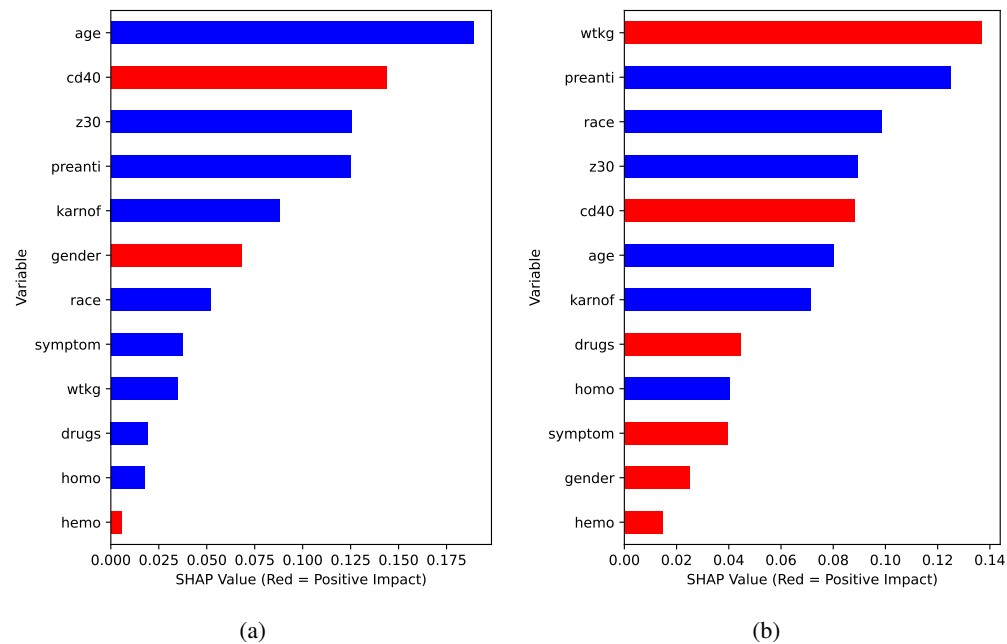

Figure 3: Visualization for the ACTG175 dataset: SHAP variable importance plots for decision rules RISE (a) and Exp (b), respectively. Covariates ($X$) are ranked by variable importance in descending order. Correlations between the feature and their SHAP value are highlighted in color. The red color means a feature is positively correlated with assigning treatment $A = 1$ and the blue indicates a negative correlation.

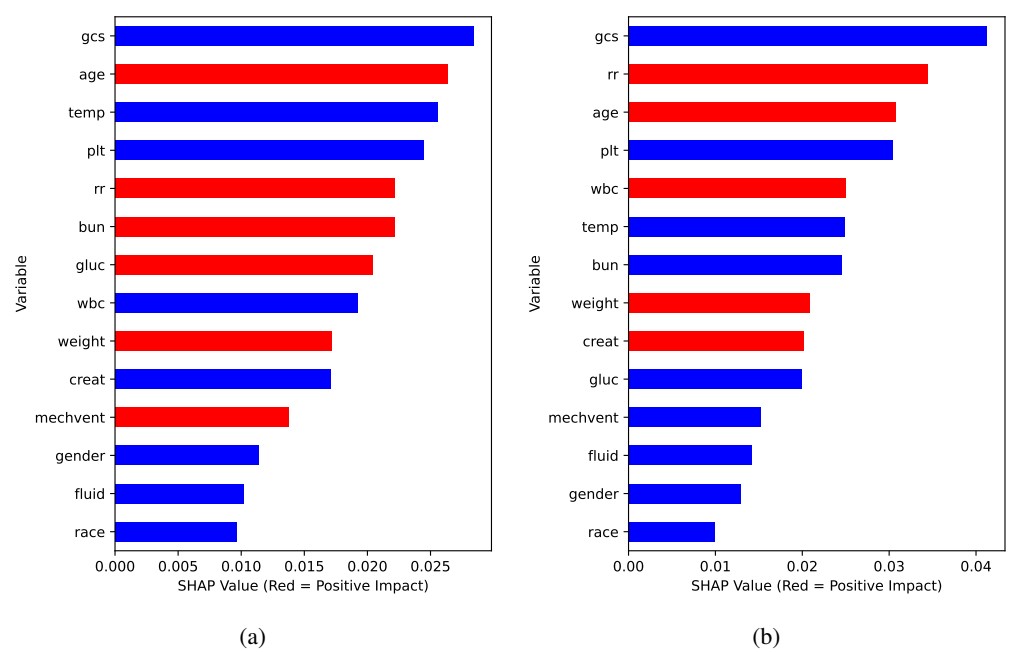

Figure 4: Visualization for the sepsis data: SHAP variable importance plots for decision rules RISE (a) and Exp (b), respectively. Covariates ($X$) are ranked by variable importance in descending order. Correlations between the feature and their SHAP value are highlighted in color. The red color means a feature is positively correlated with assigning treatment $A = 1$ and the blue indicates a negative correlation.