# OpenReview forum: "RISE: Robust Individualized Decision Learning with Sensitive Variables"
_NeurIPS.cc/2022/Conference — NeurIPS 2022 Accept_

### Official Review · Reviewer_XrZR · 2022-06-18

**Rating:** 5
**Confidence:** 3
**Soundness:** 2 fair
**Presentation:** 3 good
**Contribution:** 2 fair

**Summary:**

This paper proposes RISE, a robust individualized learning framework that improves the worst-case outcomes of decision when sensitive variables are unknown in test time, thus reducing the variation across subjects. The robustness considers the uncertainty of outcome distribution due to sensitive variables and focuses on the protection to subjects in the lower tail of the outcome distribution. A corresponding learning objective is theoretically defined and is transformed to an equivalent weighted classification problem for optimization. Experiments on real-world examples show good performance of the method and its importance from fairness and safety perspectives.

**Questions:**

1. Should the framework of RISE be regarded as a generalized form of individualized decision rules rather than a robust one? The proposed objective considering quantile- or infimum-optimal is not a must for the derivation. In other specific scenarios, for example, the venture capital investment, the highest reward might be the concern, which can also be included in this framework.
2. Is there any error between the simulation results shown in Table 2 and its theoretical results? The qualitative conclusion is clear, while the quantitative results are confusing. For example, does the numeral result 14.4 corresponds to a theoretical value of (30+0+5+27)/4=15.5?
3. Does the framework in this article refer to some works in section 'Robustness in IDRs'? All these methods are summarized with only one sentence in line 109-111, which is not informative enough.
4. The objective of RISE seems to be a protective weight increase of vulnerable groups, just like the false negative punishment in classical medical statistics to avoid serious consequences of undetected diseases. Therefore, this problem is obviously a weighted classification problem with an adaptive weight according to the objective function. Thus, the framework seems to be another paraphrase of a classic problem. Are there any other differences to be considered?
5. Have you tried to optimize the objective function in (2) directly with a neural network? It does not seem to pose a challenge. The transformation to classification problem may only serves for traditional approaches as SVM or decision trees.
6. Are there no other suitable baselines for your experiments? If so, the objective function of some other methods should be modified for your proposed task as comparison.

**Limitations:**

The discussion on limitations is sufficient.

**Strengths And Weaknesses:**

### Strengths:
1. This paper proposes a new perspective on measuring individualized decision with fairness or safety concerns, which can be extremely useful in many practical scenarios.
2. The theoretical modeling and derivation are very clear and detailed.
3. The paper is well organized and well written.

### Weaknesses
1. The related work on Robustness in individualized decision rules is insufficient, so it is hard to judge the innovation of this paper in terms of robustness learning, because similar treatment regimes as quantile-optimal have already widely used in various cases.
2. This proposed robust learning objective is essentially a minor modification to change the expectation $E_{s|x}$ into another measure, $G_{s|x}$ , which is rather intuitive with no technical innovations.
3. The derivation of the transformation from the raw problem to a weighted classification problem is not of great significance for practical training. As the optimization object is rather simple, it is reasonable to believe that neural networks can directly handle the original one.
4. The comparison in the experiments is inadequate with only vanilla baselines provided. For a fair comparison, more modern methods should be included with appropriate adjustment on their objective function to fit the proposed learning objective in this paper.

---

> ### Author Response · Authors · 2022-08-02
> **Response to Reviewer XrZR 1/2**
>
> We thank the reviewer for the thorough and detailed review of our submission. We have revised our submission significantly based on the reviewer’s questions and suggestions, marked in blue. All typos have been corrected as well. We respond to your concerns and questions one by one below.
>
> > **The related work on Robustness in individualized decision rules is insufficient, so it is hard to judge the innovation of this paper in terms of robustness learning, because similar treatment regimes as quantile-optimal have already widely used in various cases.**
>
> Thanks for the suggestions! In the rebuttal revision, we have included a detailed discussion of multiple related works related to “Robustness in IDRs” in Section 2 and highlighted their differences with our work.
>
> Below we summarize the key differences. The existing robust approaches for the decision making such as the quantile-optimal ones mainly consider robustifying the impact of residuals on the final outcome (i.e., robustifying the objective function directly), so as to avoid extreme loss during the decision making. Taking a different perspective, our work mainly focuses on robustifying the impact of sensitive variables on the outcome, which is thus substantially different from the aforementioned settings. Our motivation comes from the uncertainty in decision making caused by sensitive variables and the existing robust approaches cannot satisfy our needs. Hence, we need to propose a new methodology for robustifying the sensitive variables in decision making for optimal treatment regimes, as opposed to using the existing robust approaches.
>
>
> > **This proposed robust learning objective is essentially a minor modification to change the expectation $E_{s|x}$ into another measure, $G_{s|x}$, which is rather intuitive with no technical innovations.**
>
> We acknowledge that this modification from $E_{s|x}$ to $G_{s|x}$ in Objective function (2) is indeed a technical handling that follows intuitive thinking. However, the modification occurs on the inside of the outer expectation, hence assessing the conditional distribution over $S$ given $X$, which makes it challenging and unique from most existing robust objectives. The robustification specifically pertains to the target sensitive variable $S$. Most robust approaches directly replace the mean objective with a quantile objective on the population space (i.e., the outer expectation), and cannot be used to target the uncertainty caused by $S$. Therefore, the proposed classification-based optimization framework is designed to help solve the specific problem under consideration that is different from most existing works. We’ve also expanded our literature review in the rebuttal revision to further point out the difference between the proposed approach and earlier approaches.
>
>
> > **Have you tried to optimize the objective function in (2) directly with a neural network? It does not seem to pose a challenge. The transformation to classification problem may only serves for traditional approaches as SVM or decision trees.**
>
> We respectfully argue that neural networks cannot be directly applied in solving the original problem (e.g., Objective (3)) because the objective function is discontinuous with respect to the decision rule $d$ due to the indicator function. Therefore, there may not exist gradients, which are required for training neural networks. By transforming into a weighted classification, we are able to implement state-of-art classification-based methods including neural network models.
>
>
> > **The comparison in the experiments is inadequate with only vanilla baselines provided. For a fair comparison, more modern methods should be included with appropriate adjustment on their objective function to fit the proposed learning objective in this paper.**
>
> Thanks for the suggestions! We’ve included the doubly robust versions of Base and Exp, respectively, by adopting Policytree (PT) [1,2], the state-of-the-art policy learning method for maximizing the expected values. The two new methods are termed PT-Base and PT-Exp in our revision. We’ve included their results in all simulations and real-data applications in the rebuttal revision. See [Table 3](https://ibb.co/7NHYZMJ) and [Table 4](https://ibb.co/RPcrq7D) as an example. We suggest the reviewer read the updated Tables 3-8 in the revised main paper and the supplementary material for detailed results.
>
> The newly added mean-optimal PT-Exp method tends to show the best improvement in terms of the overall value. However, PT-Exp is shown to have minimal benefits for vulnerable subjects, which we are mainly focused on. The proposed RISE still shows the largest gain in improving the objective and value among vulnerable subjects among all compared methods.

---

> > ### Author Response · Authors · 2022-08-02
> > **Response to Reviewer XrZR 2/2**
> >
> > > **Should the framework of RISE be regarded as a generalized form of individualized decision rules rather than a robust one? The proposed objective considering quantile- or infimum-optimal is not a must for the derivation. In other specific scenarios, for example, the venture capital investment, the highest reward might be the concern, which can also be included in this framework.**
> >
> > Thanks for pointing this out. The choice of $G_{S|X}$ depends on the risk preference in decision making. As the reviewer commented, it could be quite flexible. Here we focus on lower quantiles and infimum in particular, with the motivation of improving worst-case scenarios caused by sensitive variables from fairness and safety perspectives. These are “risk-averse” settings. We do agree that “risk-seeking” settings can be fit in by considering the upper quantiles. However, considering that the contributions (motivation, theory, realization designs, real-data applications) of the paper already make the paper too crowded in terms of page limitation, we hope to focus on the already presented settings and leave the extended scope for future work where the proposed method could consider other risk preferences in different applications.
> >
> >
> > > **Is there any error between the simulation results shown in Table 2 and its theoretical results? The qualitative conclusion is clear, while the quantitative results are confusing. For example, does the numeral result 14.4 corresponds to a theoretical value of (30+0+5+27)/4=15.5?**
> >
> > Thanks for your careful reading! Yes, 14.4 is based on the numerical results of Exp. We agree that a theoretical value would be more appropriate to present in Table 2. Hence the numbers in [Table 2](https://ibb.co/YXHKVcN) for the “mean-optimal rule” have been updated to 15.5 (average of 30, 0, 5, and 27) for all and 2.5 (average of 0 and 5) for vulnerable subjects. With the newly added competitive method PT-Exp, a doubly robust version of Exp modified based on Policytree (PT) [1,2], the state-of-the-art policy learning method for maximizing the expected values, our results for Example 1 with a discrete $S$ in the updated [Table 3](https://ibb.co/7NHYZMJ) show that the numerical values of PT-Exp are close to the theoretical ones.
> >
> >
> > > **The objective of RISE seems to be a protective weight increase of vulnerable groups, just like the false negative punishment in classical medical statistics to avoid serious consequences of undetected diseases. Therefore, this problem is obviously a weighted classification problem with an adaptive weight according to the objective function. Thus, the framework seems to be another paraphrase of a classic problem. Are there any other differences to be considered?**
> >
> > We would like to point out the differences between the proposed framework and a classic weighted classification.
> > In a classification problem under the medical setting mentioned by the reviewer, the labels are known in advance and the disease group is up-weighted by some predetermined value to trade off false positives for false negatives, or vice versa. In contrast, our work focuses on addressing the uncertainty in decision making due to the sensitive variables that are not available at the time of decision making. The problem we are trying to solve is not a traditional classification problem by design. Specifically, the true labels (i.e., optimal treatment assignment) in our classification problem are unknown in advance. We need to first identify the labels by estimating the potential outcomes of different decisions on a patient, which needs to be unbiasedly determined via causal estimands. Additionally, the weights are guided by the potential difference in the gain and loss of choosing one treatment over another, which are not arbitrarily set to balance performance. Rather, the weights are used to ensure the designed reward structure is maximized.
> >
> > **References:**
> >
> > [1] Erik Sverdrup, et al. Policytree: Policy learning via doubly robust empirical welfare maximization over trees. Journal of Open Source Software, 5(50):2232, 2020.
> >
> > [2] Susan Athey and Stefan Wager. Policy learning with observational data. Econometrica, 89(1):133–161, 2021.
> >
> >
> > ***
> > ***
> >
> > We thank the reviewer once more for their detailed feedback, which helps us revise the paper greatly. We kindly ask the reviewer to consider raising their score if their concerns were appropriately addressed.

---

> > > ### Comment · Reviewer_XrZR · 2022-08-08
> > > **Raising score**
> > >
> > > Thank you for your response, and the revised part raised the integrity and persuasiveness of the whole work. I agree that this work focuses on an important issue and formulize it in a novel and rigorous way, while the solution is still a bit straightforward. Therefore, I will raise my score from 4 to 5.

---

> > > > ### Author Response · Authors · 2022-08-08
> > > > **Reminder on rating**
> > > >
> > > > Many thanks for your valuable and constructive comments on clarifying, correcting, and improving the materials in the paper!
> > > > As you have said,
> > > > > Therefore, I will raise my score from 4 to 5.
> > > >
> > > > We really appreciate your recognition. As the rebuttal DDL is approaching, could you please increase the rating in your original comment in the system? Thanks!
> > > >
> > > > We would also like to reiterate that the original objective is difficult to solve, while the proposed work manages to provide a straightforward and flexible optimization framework that can easily leverage most existing classification tools, or allows the use of off-the-shelf software packages catered to users’ needs.

---

### Official Review · Reviewer_Git7 · 2022-07-10

**Rating:** 6
**Confidence:** 4
**Soundness:** 3 good
**Presentation:** 3 good
**Contribution:** 4 excellent

**Summary:**

This paper proposes a method for robust decision-making when there is some important attribute that we have at training time, but not test time. Using a potential outcomes framework, they discuss an robust objective which aims to output good decisions for a range of values of the unavailable attribute. Studies with simulated and real data support that this method improves on "vulnerable" subjects.

**Questions:**

- seems fairly close in concept to Makar et al. "Causally Motivated Shortcut Removal Using Auxiliary Labels". Would be good to see some engagement with that work and comparison to it in terms of what each paper is aiming for (at least in the rebuttal)
- language of "sensitive" and "vulnerable" don't align that closely IMO with the algorithm proposed - I'd either target that more specifically in experiments or change the language
- how is the vulnerable group calculated in Table 4, and why is this a useful group to look at for evaluating our algorithms?
- what if we don't observe many values of S for regions of X? Are we relying mostly on the extrapolation ability of our Y-hat model? Could this create problems?

**Limitations:**

no suggestions

**Strengths And Weaknesses:**

Strengths:
- This is a useful idea and not one that I think I've seen directly proposed before
- pretty clear exposition throughout section 3 of the idea
- I appreciate the datasets in section 4.2 - seems like the authors used pretty interesting, challenging data for this evaluation

Weaknesses:
- language around "sensitive" attribute seems a little general. I understand why it's used in some sense, but really it's just any attribute we won't have access to at deployment time - as shown in section 4, this could be a whole range of things which are not really aligned which what we consider "sensitive" information
- similarly, I would say that the "vulnerable" language doesn't really align with what I would consider a vulnerable population. Rather, it's a post-hoc definition of who the standard mean-outcome model doesn't work for
-for this reason, I find the results in Table 4 a little bit hard to interpret. The paper post-hoc defines who is in our vulnerable group as exactly those users that the baseline didn't work for, and now shows that the proposed method improves on them. This would be more convincing if it was a coherent group that we saw improvement on, as we could just as easily construct such a group with the reverse property (that the baseline performs better on). Unless I'm misunderstanding this!
- unclear about how estimation works of the quantile objective when X and S are correlated - what if we don't observe many values of S for   regions of X? Are we relying mostly on the extrapolation ability of our Y-hat model?
-it's a little odd that in Table 4 RISE is best on the whole data, makes me think the results may be fairly noisy
- re: novelty, there is some related work I mention in the "Questions" section which may make the statement "we are among the first to propose a robust-type fairness criterion under causal inference" a little less true - although depends on exactly what those terms mean here

Minor points:
- line 5: it's not really convention in fair ML to ignore sensitive variables anymore
- line 201: why would negativity make (3) hard to solve?
- line 204: clarify you're minimizing over f, and its range
- is the job training data from an RCT?
-line 350: rational -> rationale

---

> ### Author Response · Authors · 2022-08-02
> **Response to Reviewer Git7 1/2**
>
> We thank the reviewer for helpful feedback and we have updated our paper accordingly, marked in blue. All typos have been corrected as well. In the following, we address the reviewer’s questions one by one.
>
> > **Language around "sensitive" attribute**
>
> We chose the language “sensitive” as many of the previous works related to safety and fairness have used similar terms such as ​​sensitive, protected, or auxiliary attributes. It serves as an umbrella term for various situations under consideration. To avoid confusion, we have provided explicit definitions at the beginning of the paper. We believe that there is no confusion about this term in the scope of the paper and hope to stick with this term. To better connect with related works, we refer to the similar terms used in other papers in the paragraph “Fairness and safety in IDRs” in Section 2.
>
>
> > **Language around “vulnerable” population**
>
> We would like to clarify that in our definition of vulnerable population does not depend on the objective functions used, but only depends on the underlying true outcome models, i.e., it is “post-hoc” only in the sense that it depends on the ground truth, but not on the methods used. We designed Example 1 to highlight the extreme case when the two different objectives (RISE and mean-optimal) favor the exact opposite subpopulations and give completely different treatment recommendations. Regardless of the methods, the vulnerable group remains those colored in red in Table 1. It is just by design that the mean-optimal rule neglects this subgroup in every situation in exchange for an overall best expected outcome. However, in all other examples (including real data experiments), the suggested treatments are largely consistent between the two methods. Because in most cases, the action to improve the vulnerable population (i.e., the worst-off) group is consistent with the action to improve the average among all groups.
>
> We also hope to clarify why the vulnerable group should not be a coherent subgroup of the population. Similar to the notion of “individualized” decision rules that may vary depending on the feature $X$, the vulnerability of subjects may differ when only partial information is available ($X$), with $S$ unknown at the time of decision making. For example, as shown in our toy example, when $X < 0.5$, those with $X < 0.5$ and $S = 1$ are vulnerable, and when $X > 0.5$, those with $X > 0.5$ and $S = 0$ are vulnerable. We remark that, in practice, it is worth further efforts to investigate the vulnerable patterns in relation to $X$, which may help to improve fairness to the individualized level.
>
>
> > **How is the vulnerable group calculated in Table 4, and why is this a useful group to look at for evaluating our algorithms?**
>
> Please see our response to the above point and see also our definition in the paragraph “Identifying vulnerable subjects” in Section 3.2, which partially address this question. We remark that the identification of the vulnerable group does not depend on the method of choice, but depends on the ground truth model.
> It is also worth noting that the “all” group in Table 4 contains both vulnerable and non-vulnerable subjects, and may not adequately highlight the gain of our method because of the inclusion of non-vulnerable subjects. This is why we choose to separately show “vulnerable” in the comparisons. We emphasize that the ultimate goal of our paper is to find a robust optimal treatment regime that can improve the worst-case scenario due to the missing of sensitive variables.
>
>
> > **It's a little odd that in Table 4 RISE is best on the whole data, makes me think the results may be fairly noisy**
>
> In theory, the mean-optimal rules should have the largest strengths in improving the overall values. For the real data, however, there could be more noise and the results are subject to data distribution and Monte Carlo error.
>
> In the revision, we’ve included the doubly robust versions of Base and Exp, respectively, by adopting Policytree (PT) [1,2], the state-of-the-art policy learning method for maximizing the expected values. The two new methods are termed PT-Base and PT-Exp in our revision. The updated [Table 4](https://ibb.co/RPcrq7D) (results of real-data applications) shows that the newly added PT-Exp method tends to have the best improvement in terms of the overall value, as the doubly robust-based estimators tend to reduce variance in value estimation. In other words, RISE is no longer the best in terms of the overall expected value, as we would expect. We suggest the reviewer read the updated [Table 4](https://ibb.co/RPcrq7D) in the revised main paper. In this updated table, the pattern in Table 4 is now consistent with what we have seen in the simulated data in [Table 3](https://ibb.co/7NHYZMJ).

---

> > ### Author Response · Authors · 2022-08-02
> > **Response to Reviewer Git7 2/2**
> >
> > > **Discussion of Makar et al. "Causally Motivated Shortcut Removal Using Auxiliary Labels".**
> >
> > Thanks for bringing this paper to our attention. The biggest difference between Makar et al. and our paper is that they focus on a **prediction settings** with the idea motivated by causal inference whereas our focus is on **causal and decision making settings**. We have included a discussion of this paper in our literature review in Section 2 and summarized the differences below.
> >
> > In particular, Makar et al. attempted to reduce the influence of auxiliary labels on prediction under distributional shift, while our paper attempts to reduce the influence of a sensitive variable on the reward function under its natural variation. The two papers share the conceptual similarity of variance reduction, but the former paper focuses on the prediction setting whereas our paper focuses on the individualized decision making setting where treatment effects are not immediately available. Additionally, the source of variation considered is slightly different. The former paper considers the uncertainty due to distributional shift, whereas our paper considers the uncertainty due to the missing of $S$ in the policy class and the distribution of potential outcome over different realizations of $S$.
> >
> > > **“What if we don't observe many values of S for regions of X? Are we relying mostly on the extrapolation ability of our Y-hat model”**
> >
> > Thanks for raising this excellent point. The proposed method relies on the correctness of our Y-hat model. To this end, we’ve used fine-tuned Neural Networks with cross-validation in our implementation, which are non-parametric methods that are robust to model misspecification. It is worth further investigation on the issue of not observing many values of $S$ for regions of $X$ in future work.
> >
> >
> > *Minor points:*
> >
> > > **line 5: it's not really convention in fair ML to ignore sensitive variables anymore**
> >
> > Thanks for your suggestion! We have changed the wording to “a naive baseline” in the abstract of the rebuttal revision.
> >
> >
> > > **line 201: why would negativity make (3) hard to solve?**
> >
> > Objective function (3) is a discontinuous function, where one can use margin representation to rewrite it as in the classification problem. When the weight is negative, technically speaking, the objective function is not semi-continuous, which is therefore even harder to solve. To avoid confusion and technical burden for readers, we decided to remove this argument in this revision.
> >
> >
> > > **line 204: clarify you're minimizing over f, and its range**
> >
> > Thanks for your close reading. We have added in the revision that we are minimizing over $f$, which is a function in the class $\mathcal{F}$, which is a class of all measurable functions. The optimization is taken over $f \in \mathcal{F}$ now.
> >
> >
> > > **“is the job training data from an RCT?”**
> >
> > Yes.
> >
> >
> > **References:**
> >
> > [1] Erik Sverdrup, et al. Policytree: Policy learning via doubly robust empirical welfare maximization over trees. Journal of Open Source Software, 5(50):2232, 2020.
> >
> > [2] Susan Athey and Stefan Wager. Policy learning with observational data. Econometrica, 89(1):133–161, 2021.
> >
> >
> > ***
> > ***
> >
> > We thank the reviewer again for supporting our work, and we kindly ask the reviewer to consider raising the score if we fully addressed the reviewer’s concern.

---

> > > ### Author Response · Authors · 2022-08-09
> > > **Looking forward to your feedback**
> > >
> > > Dear Reviewer Git7, Thank you for your invaluable feedback. Given the closing window of the rebuttal period, we were wondering whether our response and the revised manuscript addressed your concerns. If you have any additional comments, please let us know, we would be happy to address them. We kindly ask you to consider raising your scores if the concerns were appropriately addressed.

---

### Official Review · Reviewer_6nvm · 2022-07-10

**Rating:** 6
**Confidence:** 3
**Soundness:** 3 good
**Presentation:** 3 good
**Contribution:** 2 fair

**Summary:**

This paper is concerned with identifying the best decision option for individuals in the presence of sensitive variables---variables that are relevant to the decisions but are not available at the time the decision is made---at inference time. Existing decision-making frameworks may simply ignore sensitive variables or seek to optimize the mean utility overall all the individuals. Such solutions might lead to bias decision-making or unfair outcome toward different subgroups in the population. To resolve these issues, the paper takes a robust approach that optimizes with respect to the best worst-case scenarios. It does so by solving a weighted classification problem. Experiments on both synthetic data and three real-world datasets are conducted to demonstrate the performance of the proposed method.

**Questions:**

* Can the authors please clarity what is the meaning of "individualized"? For example, in equation 2, there is an expectation over $X$, which seems to indicate the average utility rather than the individualized utility?
* Related to this, in line 174-175, the author also mentioned that the decision rule is only related to $X$. Does that mean the decision rule is not related to $S$? If so, it is a bit counter-intuitive that it is still necessary to factor in $S$ in decision-making.

**Limitations:**

The paper discussed limitations. It would also be desirable to discuss whether it is possible to further relax some of the assumptions that the paper is based on (e.g. unconfoundedness).

**Strengths And Weaknesses:**

Strengths:
* The paper is concerned with an interesting and important problem in decision making.
* The proposed approach that optimizes with respect to the best worst-case scenarios makes sense, and the paper also offers a conversion of the problem into solving weighted classification problems.
* The paper is clearly written overall with intuitive explanations and examples provided.

Weakness:
* While the paper offers an empirical evaluation of the proposed method over both synthetic and real-world datasets. It is not clear to me whether the competing approaches that the proposed method compared to are competitive, representative, and exhaustive.
* The method also operates under the unconfoundedness assumption, which might not be easy to satisfy in practice. It will be desirable to carry out further synthetic experiments to understand the robustness of the proposed approach when various assumptions of the proposed approach is violated.

---

> ### Author Response · Authors · 2022-08-02
> **Response to Reviewer 6nvm**
>
> We thank the reviewer for the helpful and suggestive review. We respond to the reviewer’s questions as follows. We have also updated the paper substantially, where the major changes are marked in blue.
>
> > **It is not clear to me whether the competing approaches that the proposed method compared to are competitive, representative, and exhaustive**
>
> Thanks for the suggestions! We have included the doubly robust versions of Base and Exp, respectively, by adopting Policytree (PT) [1,2], the state-of-the-art policy learning method for maximizing the expected values. The two new methods are termed PT-Base and PT-Exp in our revision. We’ve included their results in all simulations and real-data applications in the rebuttal revision. See [Table 3](https://ibb.co/7NHYZMJ) and [Table 4](https://ibb.co/RPcrq7D) as an example. We suggest the reviewer read the updated Tables 3-8 in the rebuttal revision for detailed results.
>
> The newly added mean-optimal PT-Exp method tends to show the best improvement in terms of the overall value. However, PT-Exp is shown to have minimal benefits for vulnerable subjects. The proposed RISE still shows the largest gain in improving the objective and value among vulnerable subjects among all compared methods.
>
> > **It will be desirable to carry out further experiments when assumptions are violated**
>
> To further test the proposed method, we’ve added two additional simulations in Appendix A.4 for the scenarios where the causal assumptions may be violated. In particular, [Table 7 and 8](https://ibb.co/3dPcSq9) show the results when positivity assumption is nearly violated, and when unconfoundedness assumption is violated, respectively. The proposed RISE still shows the largest gain in improving the objective and value among vulnerable subjects among all compared methods while maintaining an overall comparable expected value. This demonstrates the robustness of our method.
>
> > **Clarification of the meaning of "individualized"? In equation 2, there is an expectation over $X$, which seems to indicate the average utility rather than the individualized utility**
>
> In our paper, $D$ is a class of decision rules that depend on $X$. Maximizing the average utility over all samples is equivalent to finding a decision rule that maximizes each individualized average outcome. To see this more clearly, by the exchange of maximization and expectation, the optimal decision rule will maximize the reward for every $X$.
>
> > **Clarification of why the train individualized decision rule is a function of $X$ only, not both $X$ and $S$**
>
> Ideally, we would hope to include both $X$ and $S$ in the decision rule. However, in the considered setting, the inclusion of $S$ in the decision making is not allowed. For example, a blood test requires time to process and its result (i.e., sensitive variable $S$) may not be available for decision making at *the time of blood draw*. Meanwhile, we do have the measurements of $S$ available in the offline training data (as $S$ is collected later), hence we can incorporate them to estimate the conditional treatment effect for assisting our decision making. If we do not adjust for the confounding effect of $S$, due to Assumptions 1b and 1c, the evaluation of any treatment regime could be biased. Then, the resulting optimal regime could be sub-optimal. This is a setting that is commonly considered in the fairness literature, although most of their works consider a prediction setting rather than a causal and decision-making setting. We also provide a causal diagram and a decision-making diagram via [link](https://ibb.co/M8QYZyD) to help better understand the idea.
>
> > **It would also be desirable to discuss how to further relax some of the assumptions, e.g. unconfoundedness**
>
> Thanks for raising this point! It is worth mentioning that our work can be naturally extended to the scenario where there exist unmeasured confounders. As long as the conditional treatment effect given observed covariates can be identified (e.g., via instrumental variables [3]), our method can be applied for finding a robust optimal decision rule. We have also updated this in the discussion section of the revision.
>
> **References:**
>
> [1] Erik Sverdrup, et al. Policytree: Policy learning via doubly robust empirical welfare maximization over trees. Journal of Open Source Software, 2020.
>
> [2] Susan Athey and Stefan Wager. Policy learning with observational data. Econometrica, 2021.
>
> [3] Linbo Wang and Eric Tchetgen Tchetgen. Bounded, efficient and multiply robust estimation of average treatment effects using instrumental variables. Journal of the Royal Statistical Society: Series B (Statistical Methodology), 2018.
>
> ***
> ***
>
> We thank the reviewer once more for their detailed feedback -- it has inspired extensive additional updates to our work and has made it considerably stronger. In light of these updates, we kindly ask the reviewer to consider raising their score if their concerns were appropriately addressed.

---

> > ### Author Response · Authors · 2022-08-09
> > **Looking forward to your feedback**
> >
> > Dear Reviewer 6nvm, We hope our reply to your final comment has reached you. Given the closing window of the rebuttal period, we were wondering whether our response and the revised manuscript addressed your concerns. We kindly ask you to consider raising your scores if the concerns were appropriately addressed.

---

### Official Review · Reviewer_ouLu · 2022-07-11

**Rating:** 6
**Confidence:** 3
**Soundness:** 3 good
**Presentation:** 3 good
**Contribution:** 3 good

**Summary:**

The authors propose a robust decision learning framework with quantile or infimum optimal treatment objective. Their idea is to improve the worst-case scenarios of individuals when decisions, with uncertainty, are required, to be made with sensitive albeit important information missing.

**Questions:**

See main review in Strengths and Weaknesses.

**Limitations:**

See main review in Strengths and Weaknesses.

**Strengths And Weaknesses:**

# Introduction

- First paragraph: I do not think this is convincing. Certainly there are sensitive variables, but a few real-world examples would help solidify your examples. They certainly crop up in the media from time to time, but which organisation is actively pursuing this design philosophy in their models? Please give real examples.
- L31: you say “The main question of interest is whether the learned IDR could yield similar outcomes across all realizations of the sensitive variables.” -  why is this the main question? I do not follow the logic as to why focus is on this.
- Table 1: you’ll need to put some work into re-jigging this table. The expectations are the coloured (and black) entries but you’ve but the expectation column-header over the action-space column. This is confusing. Further, the whole example is somewhat confusing. Attack your example with the assumption that your reader is not an expert in your field and so start from first-principles, be verbose, and explain like you would to a novice.

# Related work

- You use a version of this comment a few times in this section: “See [your listed references] and references therein for a comprehensive review”. That is not helpful. Give us a summary of what those sources say rather than divert your reader to another suite of papers. You are meant to give us a micro-review herein as this is the related work section, that you have chosen to include (you don’t have to). Now you are just giving us a more reading to do. This is a conference paper, not a journal paper, so you are forced to compress and combine. Right now what you are doing is not helpful to our (the reader’s) understanding of the state-of-the-art in this area.

# Robust Decision Learning Framework with Sensitive Variables

- Equation 2: why is $d^*$ not equal to the right-hand side but lies in the set of optimal IDRs? Does equation (2) yield one optimal IDR or multiple? If it is the former then $\in$ should be an equality instead? If the latter, please explain further how the argmax yields multiple optimal IDRs. Algorithm 1 seems to suggest that one IDR is output by RISE.
- You appear to have dropped $d \in \mathbb{D}$ at the bottom of page four, in the aromas.
- What is the function of $n^{-1}$ in equation 3? Or rather, why the inclusion of a weighting?
- Algorithm 1 would be more elegant if you looped over n rather than saying “for every i”.

# Numerical studies

- Line 250: it would appear that you’re missing the formatting for the indicator function in your estimator for $V(d)$.
- It would be helpful, at least for the synthetic experiments, to visualise the underlying causal structure and if possible, how it fits in the IDR - the causal diagram is a key-part of causal inference so it would be interesting to see how you graphically include it (if possible).
- Line 309: what on earth is a “causal forest” your reference title, [45], says ‘random forest’ but you must be referring to something else, hence what is a causal forest? Is there a reference?
- Line 328: again causal forest - reference?
- Line 341: explain why “lactate and Sequential Organ Failure Assessment (SOFA) score 4 hours before sepsis onset” are sensitive variables?

# Discussion

- Line 372, typo: “find” —> “finding”

---

> ### Author Response · Authors · 2022-08-02
> **Response to Introduction Part for Reviewer ouLu 1/3**
>
> We thank the reviewer for detailed and instructive feedback, which helps us improve the submitted manuscript greatly (marked in blue). We address the reviewer’s questions in the following one by one.
>
> > **Certainly there are sensitive variables, but a few real-world examples would help solidify your examples.**
>
> The introduction section intends to introduce the concept of sensitive variables from aspects of both delayed availability and fairness concerns followed by our proposed algorithm. We respectfully argue that delayed availability of lab measurements is a typical and inherent phenomenon in clinical settings, although there are limited existing works to address this issue. For example, in the public MIMIC-III data [1], a database of electronic health records, the timing of lab measurements is often identified by a “collection time” and a “exam time”, implying that there exists a gap between the sample collection time and the result time for lab measurements. Hence, lab measurements cannot be made available for decision making at the *collection time*.
>
> From a fairness perspective, the consideration of fairness and safety in machine learning has seen an explosion of interest in the past few years, although most of them focus on prediction, instead of a causal and decision making setting. Fairness in classification, e.g., whether admitted to a university, aims to prevent discrimination against individuals based on their membership in some group, while maintaining utility for the classifier (e.g., the university) [2]. To further demonstrate the importance and potential of our work in protecting sensitive variables in the decision making, we provide three real-world examples in Section 4.2.
>
> **References:**
>
> [1] Johnson, Alistair EW, et al. MIMIC-III, a freely accessible critical care database. Scientific data, 2016.
>
> [2] Cynthia Dwork, et al. Fairness through awareness. In Proceedings of the 3rd Innovations in Theoretical Computer Science Conference, 2012.
>
>
> > **Clarification for the sentence “The main question of interest is whether the learned IDR could yield similar outcomes across all realizations of the sensitive variables.”**
>
> Sorry for the confusion. To improve the logic flow, we have removed this sentence in the revision to avoid confusion. Our original intention was to first introduce the mean-optimal rules that assign treatments that maximize the overall outcome by averaging out the effect of sensitive variables, and then introduce the improved version (i.e., our method) that considers worst cases (as described by this sentence). We clarify that the mean-optimal method has no control over the disparity in sensitive variables. Subjects with different sensitive values may report large outcome differences, hence unfairly or unsafely treated. On the other hand, our paper focuses on improving the worst off situations caused by sensitive variables. Therefore, objective functions with robustness guarantees for sensitive variables are preferred, since they offer protection to subjects in the lower tail of the outcome distribution with regards to the sensitive values. That is, assigning a treatment that yields a more similar outcome across all realizations of the sensitive variables.
>
>
> > **Readability of Table 1**
>
> Thank you for your suggestions! We have added more descriptions in the main text including a footnote of the toy example and updated [Table 1](https://ibb.co/xYyNFFr) based on your suggestions in the rebuttal revision.
>
> Here we provide a more detailed explanation for the example. As mentioned previously in the third paragraph in Section 1, the mean-optimal rule aims to assign a treatment that has the largest average outcome given $X$ averaging out the effects of $S$, as $S$ is not available at the time of decision. For example, limiting to $X \leq 0.5$, for $A=-1$, the average outcome across $S$ group is $(11+13)/2=12$; and for $A=1$, the average outcome is $(30+0)/2=15$. As $15>12$, the mean-optimal rule assigns $A=1$ instead of $A=-1$. However, assigning $A=1$ could result in great harm for subjects who turn out to have an $S=1$, as they could get the worst outcome of $0$ if given $A=1$, highlighted in red. On the contrary, the proposed RISE tries to improve this worst-case outcome by assigning $A=-1$, So the vulnerable subjects will have an expected outcome of $13$ instead of $0$, although among all subjects with $X \leq 0.5$, the overall outcome is $12$,  which is smaller compared to $15$ by using the mean-optimal rule. This can be regarded as a trade-off for protecting the worst-case outcome induced by omitting sensitive variables in the decision making. This is similar for $X > 0.5$. Because the average of $5$ and $27$ is $16$, and the average of $15$ and $13$ is $14$. $16$ is larger than $14$, so the mean-optimal rule recommends $A=-1$. However, the proposed RISE assigns treatment ($A=1$) in order to protect those with an $S=0$ from getting a much worse outcome $5$.

---

> > ### Author Response · Authors · 2022-08-02
> > **Response to Related Work and Method Part for Reviewer ouLu 2/3**
> >
> > > **[Related Work] “Not enough description”: Give us a summary of what those sources say rather than divert your reader to another suite of papers.**
> >
> > Thank you for helping us improve the readability of the paper. Previously we put an additional literature review in Appendix A.1 because of page limitation. In the updated rebuttal version, we have added a more detailed and dedicated discussion of several related works to Section 2. Specifically, we have introduced more methods of different categories on general IDRs under causal settings, and we have also included a discussion of multiple related works in the “Robustness in IDRs” paragraph and highlighted their differences with our work.
> >
> > Below we summarize the key differences between works on “Robustness in IDRs” and our method. The existing robust approaches for the decision making such as the quantile-optimal ones mainly consider robustifying the impact of residuals on the final outcome (i.e., robustifying the objective function directly), so as to avoid extreme loss during the decision making. Taking a different perspective, our work mainly focuses on robustifying the impact of sensitive variables on the outcome, which is thus substantially different from the aforementioned settings. Our motivation comes from the uncertainty in decision making caused by sensitive variables and the existing robust approaches cannot satisfy our needs. Hence, we need to propose a new methodology for robustifying the sensitive variables in decision making for optimal treatment regimes, as opposed to using the existing robust approaches.
> >
> >
> > > **[Method] “One optimal IDR or multiple”: Equation 2: why is $d\*$ not equal to the right-hand side but lies in the set of optimal IDRs? Does equation (2) yield one optimal IDR or multiple?**
> >
> > Theoretically, there could be cases when $E[Y | X = x, A = 1] = E[Y | X = x, A =-1]$ for some $x$, which indicates that any treatment will be optimal, i.e., a tie. Therefore, there may exist multiple optimal treatment regimes. Based on this argument, Equation (2) could yield multiple solutions, hence the notation $\in$ is used for technical accuracy. From an algorithmic perspective, it is sufficient to output one of the IDRs in this set for decision making because they all share the same optimal value. In Algorithm 1 of Section 3.3, we have added a footnote in the rebuttal revision discussing the strategy when a tie situation happens. In this situation, either treatment or control could be assigned.
> >
> >
> > > **[Method] “What is the function of $n^{−1}$ in equation 3? Or rather, why the inclusion of a weighting?”**
> >
> > This equation is a sample version of the expectation $E_X \{ \mathbb{1}(d(X) = 1) [G_{S|X} \{E (Y|X,S,A=1) \}- G_{S|X} \{E (Y|X,S,A=-1) \}] \}$ in Proposition 1, hence we have the $n^{−1}$. Technically, removing it will not affect the optimization as it is a constant. We keep it here to stay consistent with the notion of expectation in Equation (2).
> >
> >
> > > **[Method] Algorithm 1 would be more elegant if you looped over n rather than saying “for every i”.**
> >
> > Thank you for your suggestion! We have updated Algorithm 1 to loop over $n$ in the revision.

---

> > > ### Author Response · Authors · 2022-08-02
> > > **Response to Numerical Studies Part for Reviewer ouLu 3/3**
> > >
> > > > **[Numerical] “Causal diagram”: It would be helpful, at least for the synthetic experiments, to visualise the underlying causal structure and if possible, how it fits in the IDR - the causal diagram is a key-part of causal inference so it would be interesting to see how you graphically include it (if possible).**
> > >
> > > We have provided a causal diagram and a decision-making diagram under consideration, respectively, via [link](https://ibb.co/M8QYZyD). The diagrams are also included in Figure 4 in Appendix A.6 in the rebuttal revision.
> > >
> > > As the causal diagram shows, both $X$ and $S$ confound the effect of treatment $A$ on outcome $Y$. The arrows represent the causal relationship between variables. Hence, both $X$ and $S$ should be used for decision making in general. On the other hand, in the decision diagram under our setting, sensitive variables $S$ are shown in a dotted circle as $S$ may not be readily available at the time of decision making. We connect $S$ and $A$ with a dotted arrow to indicate that $S$ is incorporated during model training of the decision rule, but not model deployment. Hence, the decision rule $d$ only maps $X$ to the treatment option, i.e., $A=d(X)$.
> > >
> > >
> > > > **[Numerical] “Causal forest”: Line 309: what on earth is a “causal forest” your reference title, [45], says ‘random forest’ but you must be referring to something else, hence what is a causal forest? Is there a reference?**
> > >
> > > Sorry for the confusion. Reference [45] proposed a random forest method for causal inference and they termed it “causal forest” in both the abstract and the main text. This method is also widely referred to by the name of “causal forest” in the field of causal inference.
> > >
> > > Reference [45]:
> > >
> > > Stefan Wager and Susan Athey. Estimation and inference of heterogeneous treatment effects using random forests. Journal of the American Statistical Association, 2018
> > >
> > >
> > > > **[Numerical] Line 341: explain why “lactate and Sequential Organ Failure Assessment (SOFA) score 4 hours before sepsis onset” are sensitive variables?**
> > >
> > > Sorry for the confusion and thanks for helping us improve the readability of the paper. Previously we put the detailed rationale in Appendix A.5 because of page limitation. In the updated rebuttal version, we have put more details back in the main paper.
> > >
> > > In particular, lactate and SOFA score have been two important indicators of sepsis severity and have been found to be more useful for predicting the outcome of sepsis than other clinical vitals and comorbidity scores [3,4,5]. Different from the baseline patient characteristics such as age and gender or common vital signs such as respiratory rate and temperature that can be obtained at the admission of patients, the SOFA score combines the performance of several organ systems in the body such as neurologic, blood, liver, and kidney [6], which requires additional calculation and cannot be obtained immediately. Lactate labs, which measure the level of lactic acid in the blood [7], are less common to collect in routine examinations, which could be also delayed in ordering. Hence, these two variables are treated as sensitive variables in our rea-data example as their information may not be available by the time of treatment decision due to multiple reasons including doctors' delayed ordering, long laboratory processing time, or the rapid deterioration of development of sepsis, which poses tremendous difficulties for early diagnosis and treatment decisions within a short time.
> > >
> > >
> > > **References:**
> > >
> > > [3] Michael D Howell, et al. Occult hypoperfusion and mortality in patients with suspected infection. Intensive Care Medicine, 2007.
> > >
> > > [4] Uma Krishna, et al. An evaluation of serial blood lactate measurement as an early predictor of shock and its outcome in patients of trauma or sepsis. Indian Journal of Critical Care Medicine, 2009.
> > >
> > > [5] Manu Shankar-Hari, et al. Developing a new definition and assessing new clinical criteria for septic shock: for the third international consensus definitions for sepsis and septic shock (sepsis-3). The Journal of the American Medical Association, 2016.
> > >
> > > [6] Christopher W Seymour, et al. Assessment of clinical criteria for sepsis: for the third international consensus definitions for sepsis and septic shock (sepsis-3). The Journal of the American Medical Association, 2016.
> > >
> > > [7] Lars W. Andersen, et al. Etiology and therapeutic approach to elevated lactate levels. Mayo Clinic Proceedings, 2013.
> > >
> > >
> > > ***
> > > ***
> > >
> > > We thank the reviewer once more for their detailed feedback, which helps us to make a strong revision. We kindly ask the reviewer to consider raising their score if their concerns were appropriately addressed.

---

> > > > ### Comment · Reviewer_ouLu · 2022-08-08
> > > > **Thanks**
> > > >
> > > > Thanks to the authors for responding to my review and their effort to do it in minute detail. Though we disagree on the prevalence of 'causal forest in the causal inference literature, I commend the authors for a good rebuttal.

---

> > > > > ### Author Response · Authors · 2022-08-08
> > > > > **Thank you for your feedback**
> > > > >
> > > > > We thank the reviewer for detailed suggestions and comments to help make our work a strong version.
> > > > >
> > > > > In the updated rebuttal revision, the description of the causal forest method has been added with more details. Please refer to Section 4.2 (Real-data Applications).
> > > > >
> > > > > We kindly ask the reviewer to consider raising their score if their concerns were appropriately addressed.

---

### Author Response · Authors · 2022-08-02
**Thanks for the Reviews and Summary of Key Paper Changes**

We are grateful to all reviewers for their insightful and positive feedback. We are encouraged that they found the submitted work to be a novel (Git7, XrZR) and interesting (6nvm, XrZR) idea, a useful and important research problem (ouLu, Git7, XrZR), with well-written intuitive examples (6nvm), theoretical derivation (Git7, XrZR) and interesting and challenging real-data applications (Git7).

We appreciate all reviewers’ constructive feedback and questions; We have revised our paper based on their suggestions. We summarize all major changes that we have made below. All changes are marked in blue in the updated main paper and supplementary materials.
- We added more detailed discussions on several related works in Section 2.
- We added two strong baselines based on Policytree [1,2], the state-of-the-art policy learning method for maximizing the expected values, for comparison. The corresponding results have been included in all simulations and real-data applications. Our main conclusions remain the same. Please see Tables 3-8 for detailed results.
- We conducted two additional simulation studies, whose results are reported in Appendix A.4, where causal assumptions are possibly violated, in order to further test the robustness of our method. Please see Tables 7-8 in the supplementary materials.

Tables and figures mentioned in our responses are provided via anonymized links for quick access. They are also available in the rebuttal revision.

**References:**

[1] Erik Sverdrup, Ayush Kanodia, Zhengyuan Zhou, Susan Athey, and Stefan Wager. Policytree: Policy learning via doubly robust empirical welfare maximization over trees. Journal of Open Source Software, 5(50):2232, 2020.

[2] Susan Athey and Stefan Wager. Policy learning with observational data. Econometrica, 89(1):133–161, 2021.

---

> ### Author Response · Authors · 2022-08-06
> **Please let us know if you have further concerns**
>
> Dear Reviewers, thank you for your invaluable feedback. We were wondering whether our response and the revised manuscript addressed your concerns. If you have any additional comments, please let us know, we would be happy to address them. We kindly ask you to consider raising your scores if the concerns were appropriately addressed.

---

### Meta-Review · Area_Chair_Z3qE · 2022-08-25

**Recommendation:** Accept
**Confidence:** Less certain

**Metareview:**

Reviewers are on the whole positive about this paper, after detailed responses from the authors. If this paper is accepted, many readers will be interested, and it does have the potential to be useful for real-world applications in medicine and elsewhere. Therefore, on balance, the paper does reach the standard required for NeurIPS.

**Award:**

No

---

### Decision · Program_Chairs · 2022-09-14

Accept